# VR VideoReasonBench: Can MLLMs Perform Vision-Centric Complex Video Reasoning?

**Yuanxin Liu**[1,2]  **Kun Ouyang**[1]  **Haoning Wu**[2*]  **Yi Liu**[1]  **Lin Sui**[2]
**Xinhao Li**[3]  **Yan Zhong**[2,4]  **Y. Charles**[2]  **Xinyu Zhou**[2†]  **Xu Sun**[1†]
[1] State Key Laboratory of Multimedia Information Processing,
School of Computer Science, Peking University
[2] Moonshot AI  [3] Nanjing University  [4] School of Mathematical Sciences, Peking University
liuyuanxin@stu.pku.edu.cn   wuhaoning@moonshot.cn

## ABSTRACT

Recent studies have shown that long chain-of-thought (CoT) reasoning can significantly enhance the performance of large language models (LLMs) on complex tasks. However, this benefit is yet to be demonstrated in the domain of video understanding, since most existing benchmarks lack the reasoning depth required to demonstrate the advantages of extended CoT chains. While recent efforts have proposed benchmarks aimed at video reasoning, the tasks are often knowledge-driven and do not rely heavily on visual content. To bridge this gap, we introduce **VideoReasonBench**, a benchmark designed to evaluate **vision-centric, complex video reasoning**. To ensure visual richness and high reasoning complexity, each video in VideoReasonBench depicts a sequence of fine-grained operations on a latent state that is only visible in part of the video. The questions evaluate three escalating levels of video reasoning skills: recalling observed visual information, inferring the content of latent states, and predicting information beyond the video. Under such task setting, models have to precisely recall multiple operations in the video, and perform step-by-step reasoning to get correct final answers for these questions. Using VideoReasonBench, we comprehensively evaluate 18 state-of-the-art multimodal LLMs (MLLMs), finding that most perform poorly on complex video reasoning—e.g., GPT-4o achieves only 6.9% accuracy—while the thinking-enhanced Gemini-2.5-Pro significantly outperforms others with 56.0% accuracy. Our investigations on "test-time scaling" further reveal that extended thinking budget, while offering none or minimal benefits on existing video benchmarks, is essential for improving the performance on VideoReasonBench. **Project Page:** https://llyx97.github.io/video_reason_bench/

## 1 INTRODUCTION

Recent advances in long chain-of-thought (CoT) reasoning (DeepSeek-AI et al., 2025; Jaech et al., 2024; Team et al., 2025a) have remarkably enhanced the problem-solving capabilities of large language models (LLMs). By scaling up the test-time compute with extended CoT reasoning chains, substantial performance gains have been observed in complex tasks such as mathematics (MAA, 2024; Lightman et al., 2024; Lu et al., 2024), coding (Jain et al., 2024; Jimenez et al., 2024), and scientific reasoning (Rein et al., 2023). However, the benefits of long CoT reasoning have not been fully demonstrated in the domain of video understanding. This gap is largely due to limitations in existing benchmarks (Mangalam et al., 2023; Fu et al., 2024; Liu et al., 2024; Li et al., 2024b; Wu et al., 2024; Zhou et al., 2024; Li et al., 2024c; Shangguan et al., 2025), which often lack the reasoning depth necessary to showcase the advantages of extended CoT chains. As shown in Figure 1,

---

[*]Project Lead
[†]Corresponding Author(s)

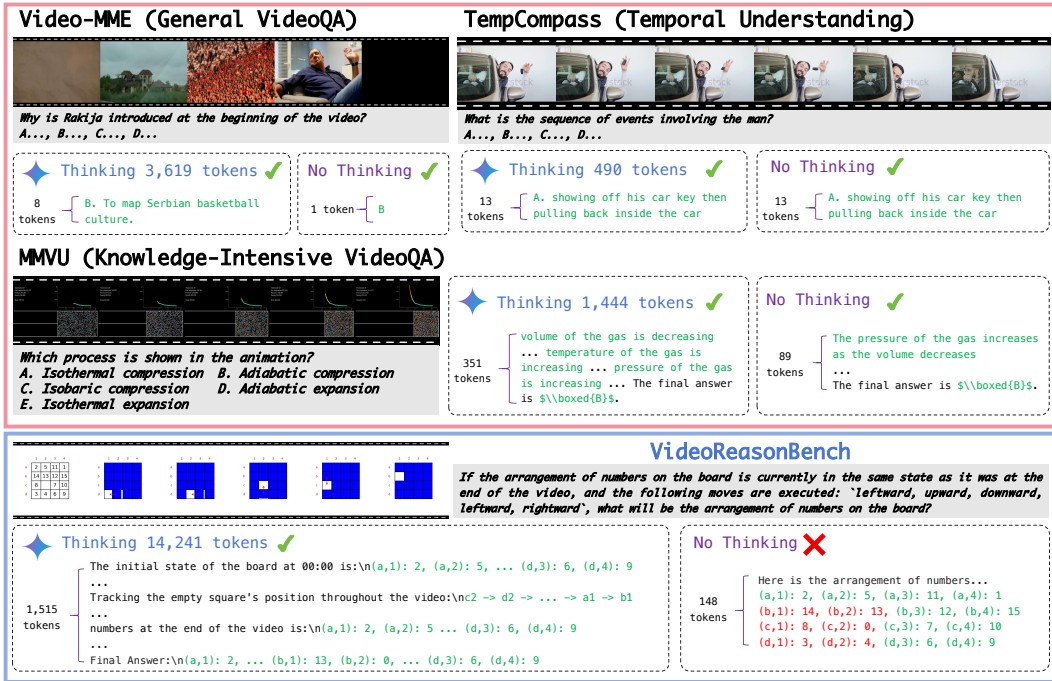

Figure 1: Examples from VIDEOREASONBENCH and three existing VideoQA benchmarks. Responses are generated by Gemini-2.5-Flash in both "Thinking" and "No Thinking" modes. The text highlighted in green/red indicate correct/incorrect responses. While questions from existing benchmarks can be answered correctly without "Thinking" using only a few tokens, VIDEORE-ASONBENCH requires "Thinking" for accurate reasoning and consumes substantially more tokens (See Figure 5 for quantitative results). It also demands finer-grained visual perception during reasoning.

the advanced multimodal LLM (MLLM) Gemini-2.5-Flash can correctly answer the questions from two popular benchmarks, Video-MME (Fu et al., 2024) and TempCompass (Liu et al., 2024) using only a few response tokens without activating the thinking mode.

To address this gap, several benchmarks have been proposed recently to better emphasize CoT reasoning in video understanding. Video-MMMU (Hu et al., 2025) and MMVU (Zhao et al., 2025) integrate video understanding with domain-specific knowledge, thus introducing a need for reasoning. However, the required reasoning process is primarily knowledge-driven, lacking a strong reliance on the visual content. Two concurrent studies, VCR-Bench (Qi et al., 2025) and MINERVA (Nagrani et al., 2025), evaluate the correctness of video reasoning process in addition to the final answer. Nonetheless, the videos and questions in these benchmarks often resemble those in general video understanding benchmarks, emphasizing short-horizon skills such as temporal grounding, action counting, and temporal order comprehension, while fall short in demanding deeper video reasoning.

Motivated by these limitations, this work introduces the **VIDEOREASONBENCH** to evaluate the capabilities of MLLMs in performing **vision-centric, complex video reasoning**. We define three levels of video reasoning, each requiring progressively more sophisticated reasoning: The **first level** is to precisely *recall* the sequential visual observations from the video. The **second level** is to *infer* latent information that is not directly observable from the video. The **third level** is to *predict* new information beyond the video. For instance, as shown in Figure 1, a video from VIDEOREASON-BENCH presents a "sliding number puzzle", in which numbered tiles are initially visible but become masked as sliding movements occur. To accurate answer the question, a model must first recall the initial tile arrangement and all subsequent movements (Level 1), then infer the final arrangement of tiles (Level 2), and finally apply this inferred information to predict future tile positions (Level 3).

VIDEOREASONBENCH is constructed based on the aforementioned core ideas. Each video illustrates a sequence of operations (e.g., sliding movements) performed to a latent state (e.g., tile ar-

rangement). The richness of visual information can be flexibly controlled by adjusting the size of the latent state and the number of operations. In addition to the "sliding number puzzle", our benchmark includes six types of video demonstrations spanning various scenes, featuring both synthetic and real-world videos. To evaluate reasoning across all three levels, we design six corresponding reasoning skills, with two for each level (see Figure 2).

Based on VIDEOREASONBENCH, we comprehensively evaluate 18 state-of-the-art MLLMs. Our results reveal that most MLLMs struggle with vision-centric complex video reasoning, achieving accuracies below 10%. In contrast, the thinking-augmented Gemini-2.5-Pro significantly outperforms all other models, reaching an accuracy of 56%. Further analysis shows that while extended chain-of-thought (CoT) reasoning offers minimal performance improvements on existing benchmarks, it is crucial to VIDEOREASONBENCH. Additionally, we observe that removing visual information from VIDEOREASONBENCH leads to a substantially larger drop in performance compared to other benchmarks, highlighting its strong reliance on visual content.

The main contributions of this work are summarized as follows:

1. We introduce the VIDEOREASONBENCH for evaluating vision-centric, complex video reasoning. It poses a necessity for models to correctly perceive multiple actions in a sequential order and perform step-by-step reasoning to finally answer the questions, therefore by-principle featuring higher demand for reasoning depth and stronger reliance on the visual content.

2. We reveal the concerning deficiency of most SOTA MLLMs in our benchmark: Several latest thinking models, such o4-mini and Seed1.5-VL, only gets around 10% accuracy; non-thinking SOTA MLLMs (e.g. GPT-4o and Qwen2.5VL-72B) scores lower than 10%; all efficient MLLMs (<10B) cannot reach even 2%.

3. Our experimental investigation confirms that the accuracy of Gemini-2.5-Flash drastically degrade while *dropping 50%* input video or *disabling thinking-mode*, while existing video benchmarks do not show similar properties. This result underscores the value of VIDEOREASONBENCH as a paragon to evaluate vision-centric complex video reasoning abilities.

## 2 VIDEOREASONBENCH

### 2.1 TASK DEFINITION

Existing research lacks a clear and established definition of what is vision-centric complex video reasoning. To address this gap, we propose a systematic framework that formally defines the task, incorporating both video content design and different reasoning question skills.

#### 2.1.1 VIDEOS

We conceptualize videos as a sequence of state transitions, represented as $\{\mathcal{S}_t, o_t, \mathcal{S}_{t+1}\}_{t=1}^{T-1}$, where an operation $o_t$ transforms state $\mathcal{S}_t$ into $\mathcal{S}_{t+1}$. In our framework, the full sequence of operations is visually observable, while the states are only partially visible—either at the beginning or at the end of the video. Thus, the visible components of a video are either: $\{\mathcal{S}_1, o_1, \cdots o_{T-1}\}$ or $\{o_1, \cdots o_{T-1}, \mathcal{S}_T\}$. This design enforces visual complexity via the dense sequence of operations and fosters reasoning complexity by requiring inference about latent states.

As illustrated in Figure 2, we design six categories of video demonstrations based on this principle. **Number:** The latent state is an $N \times N$ board with numbered tiles and one empty tile. Operations consist of sliding a numbered tile into the empty space. **Circle:** The latent state is an $N \times N$ grid containing black and white pieces. A red circle moves across the grid, flipping the color of the pieces it passes over, as well as their neighbors. **Cup:** The latent state is an $N \times N$ board with squares that may be empty or contain a coin, all covered by cups. Operations involve swapping the positions of two cups, altering the contents beneath. **File:** The latent state consists of $N$ file paths. Operations include creating, deleting, copying, and moving files within/between these paths. **Card:** The latent state comprises $N$ piles of cards. Operations involve adding a card to the top of a pile or removing a card from the bottom. **Chip:** The latent state consists of $N$ cups, each containing a number of chips. Operations involve adding or removing a chip from a cup.

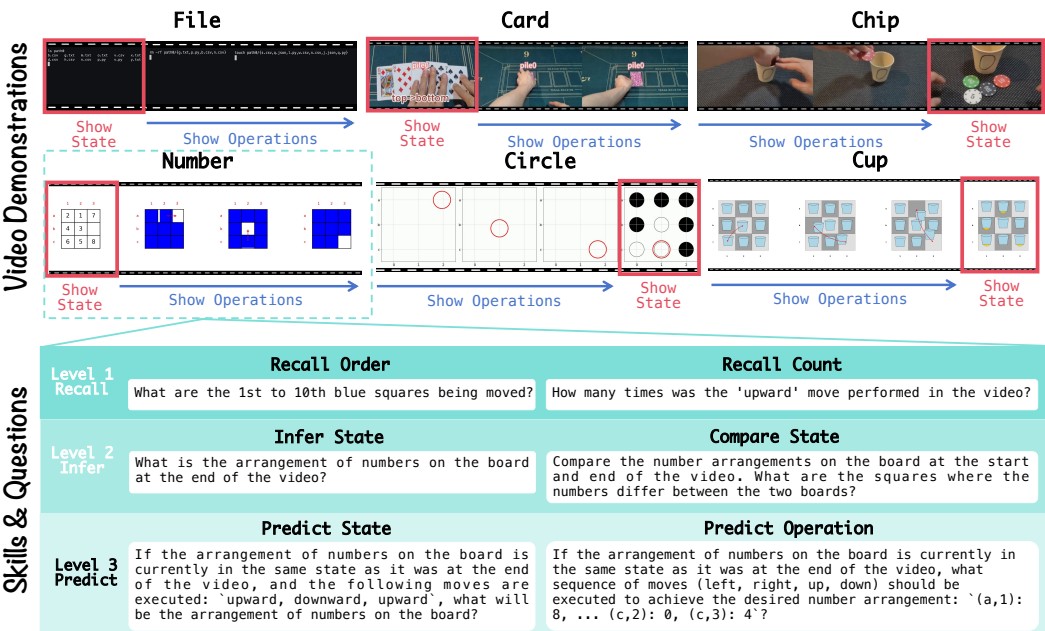

Figure 2: Illustration of vision-centric complex video reasoning. **Upper:** In each video, the latent state is revealed either at the begin or the end, and a sequence of observable operations is applied to this state. There are six categories of videos, each featuring a different type of demonstration. **Lower:** The questions assess video reasoning across three levels, with two skills for each level.

### 2.1.2 QUESTIONS

As Figure 2 shows, our framework evaluates video reasoning skills across three progressive levels.

- **Level 1** focuses on fine-grained visual perception, comprising two sub-tasks: *Recall Order*, which requires recalling the exact sequence of operations, and *Recall Count*, which involves counting the frequency of specific operations.

- **Level 2** assesses reasoning about latent states based on the observed operations. This includes *Infer State*, where the task is to infer the content of latent state at a certain moment, and *Compare State*, which requires comparing the latent state between two moments.

- **Level 3** advances to counterfactual reasoning, requiring prediction based on inferred information. It includes *Predict State*, where the goal is to predict a future state after a sequence of operations, and *Predict Operation*, which involves identifying the operations needed to reach a given target state.

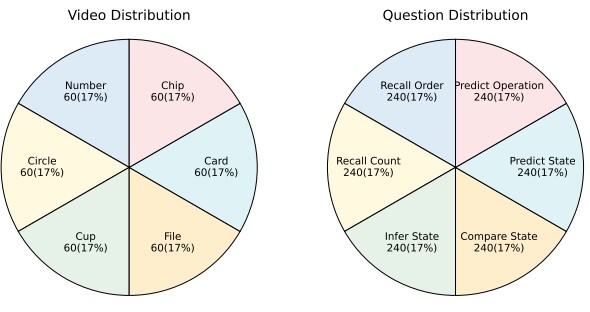

Figure 4: VIDEOREASONBENCH video and question distributions.

Table 1: VIDEOREASONBENCH statistics.

| Statistics | Value |
|---|---|
| Question Word Count (avg/max) | 81.5/262 |
| Full Prompt Word Count (avg/max) | 198.8/420 |
| Duration (Seconds, avg/max) | 54.3/154.8 |
| Videos by Operation Count | |
| $5 \sim 9$ | 120 |
| $10 \sim 14$ | 120 |
| Videos by State Size | |
| 1 & $(3 \times 3)$ | 120 |
| 2 & $(4 \times 4)$ | 120 |

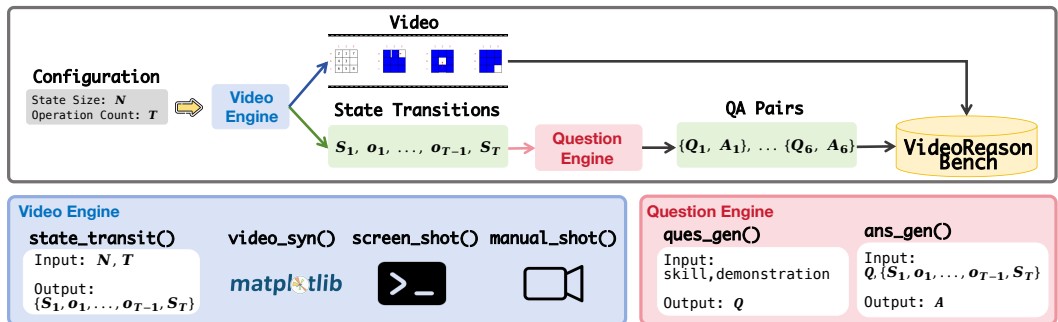

Figure 3: Overview of our data construction framework. The video engine generates state transitions from a given configuration, producing videos via Matplotlib, command-line screenshots, or real-world manual recordings. The question engine then generates questions and derives answers based on the state transitions, following the rules of each demonstration.

## 2.2 DATA CONSTRUCTION

We construct our dataset based on the previously described task definitions. Table 1 and Figure 4 summarizes the dataset statistics. In total, VIDEOREASONBENCH consists of 1,440 questions and 240 videos, with an equal number of questions per skill and an equal number of videos per demonstration. The number of operations depicted in each video ranges from 5 to 14. For demonstrations involving *File*, *Card*, and *Chip*, the latent states consist of one or two sets (e.g., one or two piles of cards), with each category containing 120 videos. For the *Number*, *Circle*, and *Cup* demonstrations, the latent states are represented as boards ranging from $3 \times 3$ to $4 \times 4$, also with 120 videos per category. To collect the dataset at scale, we design a semi-automatic data construction framework comprising two main components: a video engine and a question engine, as demonstrated in Figure 3.

### 2.2.1 VIDEO ENGINE

Given the state size $N$ and the number of operations $T$, the video engine begins by randomly initializing a state, such as an integer matrix $\mathcal{S} \in \mathbb{Z}^{N \times N}$ for the *Number* demonstration. It then generates a sequence of operations and corresponding state transitions, represented as $\{\mathcal{S}_t, o_t, \mathcal{S}_{t+1}\}_{t=1}^{T-1}$, according to predefined rules specific to each demonstration type. These transitions form a "script" that guides the video construction process. For demonstrations such as *Number*, *Circle*, and *Cup*, videos are generated programmatically using the Python Matplotlib library[1]. In the *File* demonstration, we simulate file operations within a command-line interface and capture screenshots. For real-world demonstrations like *Card* and *Chip*, we record actual videos manually.

### 2.2.2 QUESTION ENGINE

For each combination of video demonstration and reasoning skill, we define specific question templates. For instance, in the *Number* demonstration with the *Infer State* skill, the corresponding template is "`What is the arrangement of numbers on the board at the` $\{timestamp\}$ `of the video?`", where $timestamp \in \{start, end\}$ based on whether the latent state is revealed at the start or end of video. To support accurate understanding of the task, each prompt also includes a detailed description of the video demonstration, clarifying the state transition rules. Additionally, we append an answer prompt—"`Provide a summary of the final answer after 'Final Answer:'`"—after the question to help extract the final answer from the model response. The complete prompts and a comprehensive list of templates is provided in Appendix B.1. Using these templates and the associated state transition data, we automatically generate answers via hand-crafted rules, enabling efficient and accurate dataset construction.

---

[1] https://matplotlib.org/

## 2.3 EVALUATION SCHEME

Except for the *Predict Operation* category, all questions in VIDEOREASONBENCH are paired with ground-truth answers. For these questions, we evaluate model responses by inputting the question, ground-truth answer, and model-generated answer into a text-only LLM, which assesses the correctness of the response. In the *Predict Operation* category, however, ground-truth answers are not provided, as multiple valid sequences of operations can achieve the given target state. Instead, we extract operations from the model response using the text-only LLM, simulate the corresponding state transitions using the same functions employed by the video generation engine, and then verify whether the resulting state matches the target state. Detailed evaluation and operation extraction prompts are provided in Appendix B.2.

## 3 EXPERIMENTS

### 3.1 EXPERIMENTAL SETUPS

**Evaluated Models.** Based on VIDEOREASONBENCH, we conduct a comprehensive evaluation of a wide range of MLLMs: Proprietary models include the advanced GPT-4o (2024-11-20) (OpenAI, 2024) and Gemini-2.0-Flash (Pichai et al., 2024), along with the latest thinking-augmented MLLMs: o4-mini (OpenAI, 2025), Seed 1.5-VL (Guo et al., 2024), and Gemini 2.5 (Flash and Pro-0506) (Google & DeepMind, 2025). Open-source models include mPLUG-Owl3 (Ye et al., 2024), MiniCPM-V 2.6 (Yao et al., 2024), MiniCPM-o 2.6 (Yao et al., 2024), Kimi-VL-A3B (Team et al., 2025b), LLaVA-OneVision (7B and 72B) (Li et al., 2024a), LLaVA-Video (7B and 72B) (Zhang et al., 2024), InternVL3 (8B and 78B) (Zhu et al., 2025), and Qwen2.5-VL (7B and 72B) (Bai et al., 2025). These models represent the current SOTA in video understanding tasks. We also introduce an additional setting for Seed1.5-VL, Gemini-2.0-Flash and Gemini-2.5-Flash, which converts key video information into textual descriptions (see Appendix C.2 for details). This baseline, denoted as "vid2txt", allows us to independently analyze visual perception and reasoning abilities.

**Implementation Details.** For proprietary models, we use official APIs to obtain model responses. For open-source models, we perform local inference using publicly available checkpoints. For evaluation, we adopt Qwen2.5-72B (Yang et al., 2024) as the judge model. To support future research, we modify the the widely used VLMEvalKit framework (Duan et al., 2024) to support evaluation on VIDEOREASONBENCH. Additional details regarding generation configurations and video processing are provided in Appendix C.1.

To evaluate human performance on VIDEOREASONBENCH, we randomly sample 240 examples from the full set of 1,440, selecting 40 examples per reasoning skill. Three of the authors independently annotate the data. Each annotator is presented with the same video and question pairs shown to the models, and provides answers in free-text format.

### 3.2 MAIN RESULTS

Table 2 presents the evaluation results of various MLLMs and a human baseline on the proposed VIDEOREASONBENCH, from which we can derive the following findings:

**Current MLLMs struggle with vision-centric complex video reasoning.** Open-source *Efficient Models* (<10B parameters) perform near random, with accuracies below 2%. Larger "Flagship Models" (72B+ active parameters), as well as GPT-4o, also also fail to make substantial progress, with performance still below 10%. Even the most recent thinking-enhanced models, such as o4-mini and Seed1.5-VL, show only modest improvements, scoring 10.7% and 11.5% respectively. In contrast, the Gemini 2.5 series demonstrates markedly stronger performance, with Gemini-2.5-Pro-0506 achieving 56% accuracy. Since Gemini-2.5 models are not not specifically optimized for our task, this result highlights a notable gap between frontier proprietary systems and current open-source MLLMs. Nonetheless, a substantial disparity remains relative to human performance, which reaches 73.8%. These findings suggest that even the most advanced MLLMs still fall short of human-level capability in the complex video reasoning tasks posed by VIDEOREASONBENCH.

Table 2: VIDEOREASONBENCH evaluation results across three levels of reasoning skills. *The human baseline was assessed on a subset of 240 examples (40 per skill), with an average response time of 223.2 seconds per example. "vid2txt" indicates replacing the video with textual context that summarizes key information of the video content.

| Model | Act. Params | Think | Level 1 | | Level 2 | | Level 3 | | Overall |
|---|---|---|---|---|---|---|---|---|---|
| | | | Recall Order | Recall Count | Infer State | Compare State | Predict State | Predict Operation | |
| Human* | - | 223.2s | 87.5 | 90.0 | 80.0 | 75.0 | 67.5 | 42.5 | 73.8 |
| **Open-source Models** | | | | | | | | | |
| *Efficient Models* | | | | | | | | | |
| mPLUG-Owl3 | 7B | ✗ | 0.0 | 0.0 | 0.0 | 0.0 | 0.0 | 0.0 | 0.0 |
| MiniCPM-V 2.6 | 8B | ✗ | 2.1 | 0.4 | 0.4 | 0.0 | 1.2 | 0.4 | 0.8 |
| MiniCPM-o 2.6 | 8B | ✗ | 1.2 | 0.4 | 0.4 | 0.8 | 1.2 | 0.4 | 0.8 |
| LLaVA-OneVision | 7B | ✗ | 0.0 | 0.0 | 0.4 | 0.0 | 0.4 | 0.8 | 0.3 |
| LLaVA-Video | 7B | ✗ | 0.0 | 0.0 | 0.0 | 0.0 | 0.0 | 0.0 | 0.0 |
| InternVL3 | 8B | ✗ | 0.4 | 0.8 | 0.0 | 0.4 | 1.7 | 0.0 | 0.6 |
| Qwen2.5-VL | 7B | ✗ | 3.8 | 0.8 | 0.4 | 0.0 | 2.1 | 0.8 | 1.3 |
| Kimi-VL-A3B | 3B | ✗ | 1.7 | 3.3 | 1.2 | 0.4 | 1.7 | 0.0 | 1.4 |
| *Flagship Models* | | | | | | | | | |
| LLaVA-OneVision | 72B | ✗ | 0.0 | 0.0 | 0.0 | 0.0 | 0.8 | 0.0 | 0.1 |
| LLaVA-Video | 72B | ✗ | 0.0 | 0.0 | 0.0 | 0.0 | 0.4 | 0.0 | 0.1 |
| InternVL3 | 78B | ✗ | 11.2 | 14.6 | 0.8 | 2.1 | 3.8 | 2.1 | 5.8 |
| Qwen2.5-VL | 72B | ✗ | 12.5 | 17.1 | 4.2 | 4.2 | 2.9 | 2.1 | 7.2 |
| **Proprietary Models** | | | | | | | | | |
| GPT-4o | - | ✗ | 14.2 | 15.8 | 4.2 | 6.2 | 0.8 | 0.0 | 6.9 |
| o4-mini | - | ✓ | 14.2 | 20.4 | 7.1 | 11.7 | 6.2 | 4.6 | 10.7 |
| Seed1.5-VL | 20B | ✓ | 24.2 | 27.1 | 3.8 | 7.9 | 3.8 | 2.1 | 11.5 |
| Gemini-2.0-Flash | - | ✗ | 18.3 | 22.5 | 6.7 | 6.7 | 5.0 | 3.3 | 10.4 |
| Gemini-2.5-Flash | - | ✗ | 22.5 | 34.2 | 19.6 | 20.4 | 8.8 | 7.1 | 18.8 |
| Gemini-2.5-Flash | - | ✓ | 44.6 | 41.7 | 27.9 | 27.1 | 13.8 | 9.6 | 27.4 |
| Gemini-2.5-Pro-0506 | - | ✓ | **69.2** | **70.4** | **63.3** | **56.7** | **42.1** | **34.6** | **56.0** |
| Qwen2.5-VL (vid2txt) | 7B | ✗ | 32.1 | 30.8 | 7.9 | 13.3 | 1.7 | 1.7 | 14.5 |
| Qwen2.5-VL (vid2txt) | 72B | ✗ | 62.5 | 50.0 | 7.5 | 7.5 | 1.2 | 5.0 | 22.3 |
| Gemini-2.0-Flash (vid2txt) | - | ✗ | 66.7 | 52.5 | 42.9 | 37.1 | 26.2 | 20.0 | 40.9 |
| Seed1.5-VL (vid2txt) | 20B | ✓ | 83.3 | **87.9** | 74.2 | 71.7 | 54.2 | 45.4 | 69.4 |
| Gemini-2.5-Flash (vid2txt) | - | ✓ | **86.7** | 82.5 | **84.2** | **75.8** | 56.2 | 47.9 | **72.2** |

**Why do most MLLMs fail?**   The poor performance of most MLLMs can be attributed to two main limitations: **(1) Insufficient fine-grained temporal perception**. Current models struggle to capture dense sequential operations in videos, as evidenced by their inability to surpass 30% accuracy even on basic "Level 1" tasks. Strikingly, when videos are replaced with textual summaries (vid2txt), performance improves substantially—for example, Seed1.5-VL rises from 11.5% to 69.4%, and Gemini-2.5-Flash increases from 27.4% to 72.2%. This stark contrast highlights fine-grained video temporal perception as a major bottleneck for current MLLMs (Liu et al., 2024; Li et al., 2024c; Shangguan et al., 2025; Li et al., 2025). **(2) Limited capacity for multi-hop, in-depth reasoning.** Thinking-enhanced MLLMs consistently outperform the none-thinking ones. For instance, Gemini-2.5-Flash improves from 18.8% to 27.4% when enabling the thinking mode. Under the "vid2txt" setting, Seed-1.5-VL and Gemini-2.5-Flash with reasoning outperform the non-thinking Gemini-2.0-Flash by nearly 30% accuracy and the Qwen2.5-VL models by nearly 50%. This highlights the importance of explicit reasoning mechanisms and extended CoT chains in tackling the complex video reasoning problems posed by VIDEOREASONBENCH. A deeper analysis of the role of thinking in different video understanding benchmarks is provided in § 3.3.1. Additionally, § 3.5 presents case studies of model errors, offering a more intuitive illustration.

**VIDEOREASONBENCH poses substantial challenges—even for humans.**   Human annotators also face significant challenges, as answering a single question requires recognizing multiple distinct operations (up to 14) within a video and accurately inferring the corresponding latent state transitions. This process is cognitively demanding and time-intensive, taking annotators an average of 223.2 seconds per question. Furthermore, a single misinterpretation could result in an incorrect final answer, which helps explain the relative low human accuracy—especially on the tasks requiring Level 3 skills.

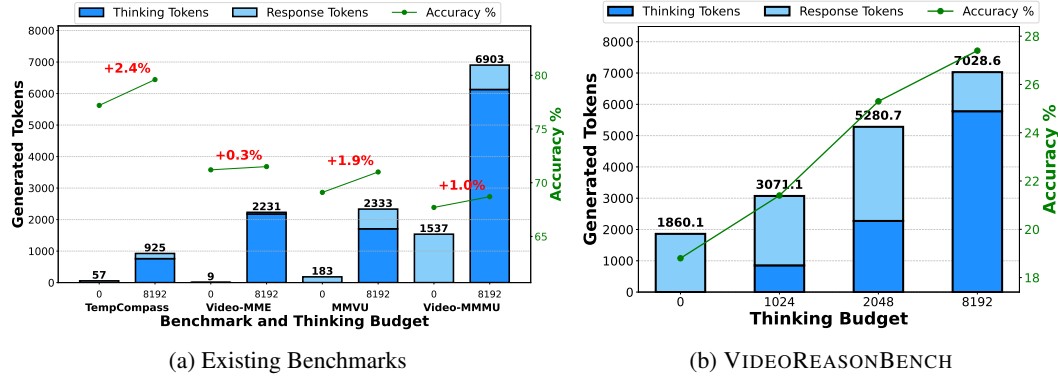

(a) Existing Benchmarks  (b) VIDEOREASONBENCH

Figure 5: Performance of Gemini-2.5-Flash with varying thinking budgets on five benchmarks. The "Generated Tokens" is the sum of "Thinking Tokens" and "Response Tokens".

Table 3: Performance of Gemini-2.5-Flash with different visual inputs on five benchmarks.

| Benchmark | Full Video | Cut 50% | Single Frame | Text-only |
|---|---|---|---|---|
| TempCompass (MCQ) (Liu et al., 2024) | 79.6 | 73.5 ($\downarrow$ 7.6%) | 59.0 ($\downarrow$ 25.9%) | 40.2 ($\downarrow$ 49.5%) |
| Video-MME (w/o subs) (Fu et al., 2024) | 71.5 | 64.1 ($\downarrow$ 10.3%) | 51.3 ($\downarrow$ 28.3%) | 45.6 ($\downarrow$ 36.2%) |
| MMVU (Zhao et al., 2025) | 70.3 | 69.1 ($\downarrow$ 1.7%) | 60.6 ($\downarrow$ 13.8%) | 44.8 ($\downarrow$ 36.3%) |
| Video-MMMU (Hu et al., 2025) | 68.7 | 64.3 ($\downarrow$ 6.4%) | 56.3 ($\downarrow$ 18.0%) | 49.7 ($\downarrow$ 27.7%) |
| VIDEOREASONBENCH (ours) | 27.4 | 12.2 ($\downarrow$ 55.5%) | 0.5 ($\downarrow$ 98.2%) | 1.0 ($\downarrow$ 96.4%) |

**Reasoning difficulty increases from Level 1 to Level 3.** Performance consistently declines from Level 1 to Level 3 reasoning skill for both MLLMs and humans. This trend strongly aligns with the intended design of the benchmark, where higher level reasoning skills are built upon lower level skills. Such design ensures an increased difficult across the three levels.

### 3.3 ANALYSIS

#### 3.3.1 EFFECT OF THINKING

The benefits of extended CoT reasoning remain underexplored in the domain of video understanding. To address this gap, we systematically investigate how varying the length of reasoning affects performance on VIDEOREASONBENCH and four representative video understanding benchmarks that focus on different abilities: TempCompass (multi-choice), Video-MME (w/o subwords), MMVU, Video-MMMU. We leverage the Gemini-2.5-Flash model, which enables explicit control over the number of reasoning tokens through a "Thinking Budget" parameter[2].

As shown in Figure 5, with the increase in thinking budget, VIDEOREASONBENCH demonstrates a notable accuracy improvement—rising by roughly 9%. In contrast, existing benchmarks exhibit minimal gains, all under 2.5%, when the same increase in thinking budget is applied. This suggests that thinking contributes more to the performance of VIDEOREASONBENCH than existing benchmarks.

Additionally, the number of response tokens varies notably across benchmarks when thinking budget is set to zero: For TempCompass and Video-MME, which primarily test basic temporal and general video understanding, responses are concise—requiring only tens of tokens. Conversely, MMVU and Video-MMU, which demand knowledge-intensive reasoning, show substantially higher response token counts, averaging 183 and 1,537 tokens respectively. Notably, VIDEOREASONBENCH produces even longer responses, averaging 1,860.1 tokens, when deprived of explicit "thinking" resources. This pattern highlights the challenging nature of VIDEOREASONBENCH.

---

[2]This parameter affects the number of thinking tokens but does not allow for precise token-level control.

Table 4: Results across different state sizes and operation counts.

| Model | State Size | | Operation Count | |
|---|---|---|---|---|
| | 1 & (3x3) | 2 & (4x4) | 5-9 | 10-14 |
| Seed1.5-VL | 11.9 | 11.0 | 15.1 | 7.8 |
| Gemini-2.0-Flash | 11.8 | 9.0 | 12.9 | 7.9 |
| Gemini-2.5-Flash | 30.4 | 24.4 | 30.1 | 24.7 |
| Gemini-2.5-Pro | 59.9 | 52.2 | 58.2 | 53.9 |

Table 5: Results across different state reveal timing.

| Model | Begin | End |
|---|---|---|
| Seed1.5-VL | 12.1 | 10.8 |
| Gemini-2.0-Flash | 13.3 | 7.5 |
| Gemini-2.5-Flash | 35.3 | 19.6 |
| Gemini-2.5-Pro | 66.9 | 45.1 |

### 3.3.2 EFFECT OF VISION RELIANCE

VIDEOREASONBENCH is designed to evaluate video reasoning that demands **fine-grained visual perception**. To assess its reliance on visual information and compare with existing benchmarks, we evaluate the performance of Gemini-2.5-Flash under four different visual input conditions: the full video, a version that randomly cuts 50% of the video, a single center frame, and a text-only input (with no visual content).

The results are shown in Table 3. For MMVU and Video-MMMU, which also involve reasoning, removing half of the video frames results in less than a 7% relative performance drop. In contrast, performance on VIDEOREASONBENCH decreases by 55% under the same condition. When the visual input is further reduced to a single frame, VIDEOREASONBENCH shows a dramatic performance decline of 98.2%, whereas the largest drop observed in the existing benchmarks is only 28.3%. These results suggest that VIDEOREASONBENCH demands a much higher degree of vision reliance than current video understanding benchmarks.

### 3.3.3 EFFECT OF VIDEO COMPLEXITY

As introduced in § 2.2, our videos vary in operation count and state size, both of which influence the richness of visual information. As we can see in Table 4, the performance of MLLMs generally decreases as operation count and state size increase. These findings indicate that video reasoning complexity can be effectively controlled by adjusting these two parameters, enabling flexible scaling of the benchmark's difficulty in future evaluations. We also include an analysis on further extending operation count and state size for *Number* and *Cup* videos in Appendix A.3.

### 3.4 EFFECT OF STATE REVEAL TIMING

In VIDEOREASONBENCH, the latent state is revealed either at the beginning or at the end of each video. Table 5 compares model performance under these two different reveal timings. As shown, all four MLLMs exhibit lower accuracy when the latent state is revealed at the end. This performance drop occurs because, in this setting, the models must infer the initial state by reasoning backward through the sequence of state transitions. This reverse inference is inherently more challenging than following the transitions in their natural, forward order.

### 3.5 CASE STUDY

To better understand why most MLLMs struggle with complex video reasoning and why Gemini-2.5-Pro excels, we conduct a case study analyzing model failure cases (Table 6). The example question requires inferring the number arrangement at the **end** of the video. However, Gemini-2.0-Flash simply outputs the visible arrangement at the **beginning**, showing a lack of reasoning. Qwen2.5-VL-72B attempts multi-hop reasoning to track state transitions but misidentifies intermediate square shifts, leading to an incorrect final answer. These cases illustrate two representative failure modes of current MLLMs in VIDEOREASONBENCH: **lack of reasoning** and **incorrect visual perception**. In contrast, Gemini-2.5-Pro successfully tracks state transitions through temporally grounded multi-hop reasoning, arriving at the correct final arrangement.

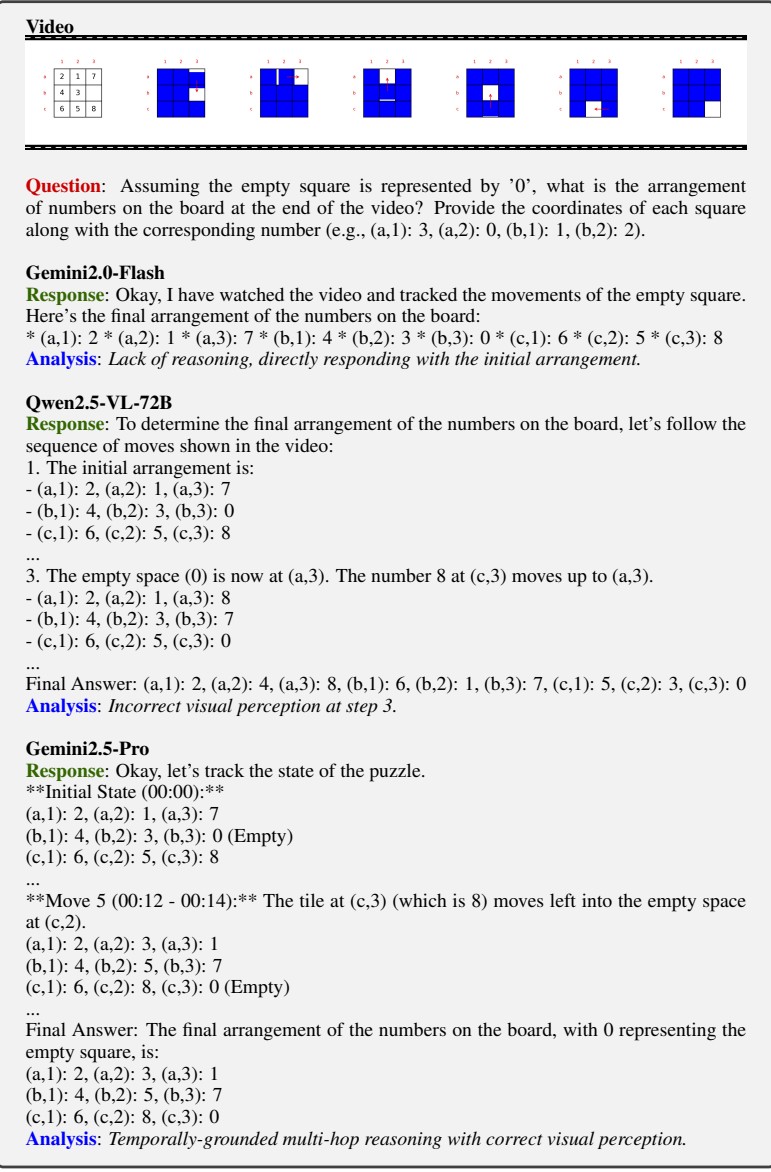

Table 6: Case study of responses by different MLLMs.

## 4 CONCLUSIONS

This paper presents VIDEOREASONBENCH to evaluate vision-centric, complex video reasoning abilities of MLLMs. Our results reveal that most SOTA MLLMs struggle with this task, achieving very low accuracies, whereas the thinking-enhanced Gemini-2.5-Pro significantly outperforms others. A detailed analysis highlights two primary bottlenecks: the lack of fine-grained temporal perception and limited multi-hop, in-depth reasoning skills. We further show that extended chain-of-thought reasoning provides little benefit on existing video understanding benchmarks but proves essential for improving performance on VIDEOREASONBENCH. Additionally, we observe that removing visual information from VIDEOREASONBENCH leads to a substantially larger drop in performance compared to other benchmarks, underscoring its strong reliance on visual content. Overall, VIDEOREASONBENCH provides a challenging and timely testbed to advance research in complex video reasoning.

## 5 ACKNOWLEDGEMENTS

We gratefully thank the anonymous reviewers for their insightful comments that substantially improved this work. This research is supported in part by Moonshot AI and National Natural Science Foundation of China (No. 92470205).

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

## A   MORE EXPERIMENTS

### A.1   ANALYSIS OF LLM JUDGEMENT

To assess the robustness of using an LLM-based judge for evaluation, we conduct two complementary analyses: (1) We paraphrase the MLLMs' answers and compare the evaluation results to that of the original answers. (2) We replace our default judge model (Qwen2.5-72B (Yang et al., 2024)) with an alternative judge (Qwen3-235B-A22B (Yang et al., 2025)) and examine the consistency between their assessments. The results are presented in Table 7 and Table 8. As we can see, the performance of MLLMs remain highly consistent across both analyses, with average accuracy differences staying within 1%. This finding indicates that the evaluation is stable with respect to both answer reformulation and judge model choice.

### A.2   CORRELATION BETWEEN VIDEOREASONBENCH AND LMARENA PERFORMANCE

Since the videos in VIDEOREASONBENCH are designed as relatively controlled and "clean" scenarios, it is important to understand whether performance on such scenarios generalizes to real-world visual reasoning. To examine this, we compare the performance of several frontier MLLMs on VIDEOREASONBENCH with their rankings on the LMArena vision leaderboard[3], where models are evaluated based on human preference votes over responses to user-submitted, real-world queries. As shown in Table 9, the results display a clear and consistent correlation. This finding suggests that strong performance on VIDEOREASONBENCH is a reliable indicator of a model's broader and more generalized visual reasoning capabilities.

### A.3   FURTHER EXTENSION OF STATE SIZE AND OPERATION COUNT

To further investigate the impact of scaling video content complexity, we extend both the state size and the operation count in the *Number* and *Cup* demonstrations using our automatic video and QA generation engine. As shown in Table 10 and Table 11, model performance decreases as either dimension grows. Notably, however, Gemini-2.5-Pro exhibits a smaller relative drop than Gemini-2.5-Flash and Seed1.5-VL, suggesting that it is more resilient to increases in video complexity. We view state size and operation count expansion as effective and convenient mechanisms for further increasing our benchmark difficulty in future iterations.

## B   MORE DETAILS OF VIDEOREASONBENCH

### B.1   QUESTION TEMPLATES

Table 12, 13, 14, 15, 16, 17 present the question templates for the six types of video demonstrations, respectively. As we can see, the full prompt consists of three components: (1) The *Task Instruction* provides a clear and precise description of the video demonstration along with the state transition rules, thereby eliminating ambiguity in task interpretation. (2) The *Question* states the query in a complete sentence and specifies the expected answer format. (3) The *Answer Prompt* instructs the models to summarize the final answer after the identifier phrase `Final Answer:`, which facilitates easy extraction of the answer from the model's response.

### B.2   EVALUATION PROMPTS

The evaluation prompt is illustrated in Table 18 and the prompts for extracting operations from the model response are shown in Table 19.

---

[3]https://lmarena.ai/leaderboard/vision

Table 7: Evaluation results with different answer formats.

| Model | Answer Form | Recall Order | Recall Count | Infer State | Compare State | Predict State | Avg |
|---|---|---|---|---|---|---|---|
| Gemini-2.5-Flash | original | 44.6 | 41.7 | 27.9 | 27.1 | 13.8 | 31.0 |
| Gemini-2.5-Flash | paraphrased | 45.4 | 44.2 | 28.8 | 27.1 | 13.8 | 31.8 |
| Gemini-2.5-Flash w/o think | original | 22.5 | 34.2 | 19.6 | 20.4 | 8.8 | 21.1 |
| Gemini-2.5-Flash w/o think | paraphrased | 22.3 | 34.5 | 20.8 | 19.5 | 10.2 | 21.2 |
| Seed1.5-VL | original | 24.2 | 27.1 | 3.8 | 7.9 | 3.8 | 13.3 |
| Seed1.5-VL | paraphrased | 24.2 | 26.7 | 3.8 | 7.9 | 3.8 | 13.3 |

Table 8: Evaluation results with different judge models.

| Model | Judge Model | Recall Order | Recall Count | Infer State | Compare State | Predict State | Avg |
|---|---|---|---|---|---|---|---|
| Gemini-2.5-Flash | Qwen2.5-72B | 44.6 | 41.7 | 27.9 | 27.1 | 13.8 | 31.0 |
| Gemini-2.5-Flash | Qwen3-235B-A22B | 45.8 | 40.8 | 27.1 | 26.2 | 14.2 | 30.8 |
| Gemini-2.5-Flash w/o think | Qwen2.5-72B | 22.5 | 34.2 | 19.6 | 20.4 | 8.8 | 21.1 |
| Gemini-2.5-Flash w/o think | Qwen3-235B-A22B | 22.5 | 32.9 | 19.6 | 20.4 | 9.6 | 21.0 |
| Seed1.5-VL | Qwen2.5-72B | 24.2 | 27.1 | 3.8 | 7.9 | 3.8 | 13.3 |
| Seed1.5-VL | Qwen3-235B-A22B | 23.8 | 25.8 | 2.1 | 6.7 | 3.8 | 12.4 |

## C  MORE DETAILS OF EXPERIMENTAL SETUPS

### C.1  INFERENCE CONFIGURATIONS

Table 23 summarizes the key inference settings for the MLLMs, including model version, input frames, generation temperature, and max tokens. Proprietary models are queried via official APIs at 1 fps with a maximum frame limit. Open-source models run on internal 80 GB GPU clusters, with videos sampled at 32 frames for efficient models and 64 for flagship models.

### C.2  VID2TXT BASELINE

Based on the ground-truth state transition sequences, we construct textual summary of key video information. The specific examples for the six types of video demonstrations are illustrated in Table 20, 21, 22.

### C.3  HUMAN ANNOTATION

Figure 6 illustrates the annotation interface used to establish the human performance baseline. Annotators are shown the video along with the complete question prompt—excluding the *Answer Prompt*—identical to what the evaluated models receive.

## D  LLM USAGE

We employ large language models (e.g., ChatGPT[4]) to assist text polishing and grammar checking.

## E  LIMITATIONS AND FUTURE WORK

We acknowledge two primary limitations of the current work. First, the videos in VIDEOREASON-BENCH are relatively "clean", and do not account for the broader perceptual challenges present in open-world scenarios, such as motion blur, occlusion, and other sources of visual ambiguity. Second, this work focuses on establishing a rigorous evaluation benchmark and does not propose concrete methods for improving MLLM performance on the targeted reasoning tasks. Looking ahead, we see two promising research directions: (1) incorporating open-world videos to further test the robustness of video reasoning under greater perceptual stress and (2) constructing scalable video reasoning data construction pipeline and devising training objectives to enhance MLLM's video reasoning capability.

---

[4]https://chatgpt.com

Table 9: Performance on VIDEOREASONBENCH and LMArena Vision Leaderboard (Nov 23, 2025).

| Model | VIDEOREASONBENCH | LMArena |
|---|---|---|
| Gemini-2.5-Pro | 56.0 | 1249 |
| Gemini-2.5-Flash | 27.4 | 1214 |
| o4-mini | 10.7 | 1202 |
| Gemini-2.0-Flash | 10.4 | 1169 |
| GPT-4o | 6.9 | 1119 |

Table 10: Results of varying state sizes for *Number* and *Cup*.

| Model | 3×3 | 4×4 | 5×5 | 6×6 |
|---|---|---|---|---|
| Seed1.5-VL | 5.4 | 6.7 | 5.8 | 3.3 |
| Gemini-2.5-Flash | 25.8 | 19.2 | 7.9 | 6.7 |
| Gemini-2.5-Pro | 70.4 | 61.7 | 59.6 | 56.7 |

Table 11: Results of varying operation counts for *Number* and *Cup*. The values in brackets denotes average video durations.

| Model | 5-9 (20.0s) | 10-14 (32.4s) | 15-19 (44.7s) | 20-24 (57.1s) | 25-29 (69.4s) |
|---|---|---|---|---|---|
| Seed1.5-VL | 10.4 | 1.7 | 0.4 | 0.4 | 0.0 |
| Gemini-2.5-Flash | 25.8 | 19.2 | 9.6 | 5.8 | 5.0 |
| Gemini-2.5-Pro | 68.8 | 63.3 | 61.2 | 55.4 | 51.7 |

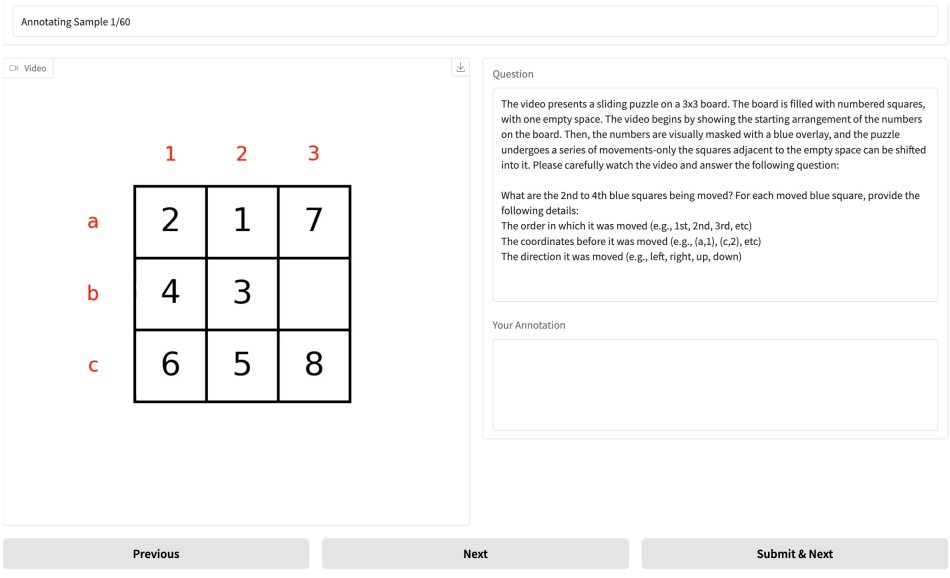

Figure 6: Screenshot of the human annotation interface.

**Video**

**Full Prompt**
{**task_instruction**}
{**question**}
{**answer_prompt**}

**Task Instruction (Show at Begin)**: The video presents a sliding puzzle on a $\{N \times N\}$ board. The board is filled with numbered squares, with one empty space. The video **begins by showing the starting arrangement** of the numbers on the board. Then, the numbers are visually masked with a blue overlay, and the puzzle undergoes a series of movements-only the squares adjacent to the empty space can be shifted into it. Please carefully watch the video and answer the following question:

**Task Instruction (Show at End)**: The video presents a sliding puzzle on a $\{N \times N\}$ board. The board is filled with numbered squares, with one empty space. Initially, all numbered squares are visually masked with a blue overlay. Then, the puzzle undergoes a series of movements-only the squares adjacent to the empty space can be shifted into it. The video **ends by showing the final arrangement** of the numbers on the board. Please carefully watch the video and answer the following question:

**Answer Prompt**: Provide a summary of the final answer after 'Final Answer:'

**Questions**
  **Recall Order**: What are the $\{start\_id\}$ to $\{start\_id\}$ blue squares being moved? For each moved blue square, provide the following details: The order in which it was moved (e.g., 1st, 2nd, 3rd, etc). The coordinates before it was moved (e.g., (a,1), (c,2), etc). The direction it was moved (e.g., left, right, up, down)

  **Recall Count**: How many times was the '$\{move\}$' move performed in the video? For each occurrence, provide the coordinate of the square (e.g., (a,1), (c,2)...) before the move.

  **Infer State**: Assuming the empty square is represented by '0', what is the arrangement of numbers on the board at the $\{timestamp\}$ of the video? Provide the coordinates of each square along with the corresponding number (e.g., (a,1): 3, (a,2): 0, (b,1): 1, (b,2): 2).

  **Compare State**: Assuming the empty square is represented by '0', compare the number arrangements on the board at the start and end of the video. What are the squares where the numbers differ between the two boards? Provide their coordinates along with the corresponding number at the $\{timestamp\}$ of the video (e.g., (a,1): 3, (b,1): 1).

  **Predict State**: If the arrangement of numbers on the board is currently in the same state as it was at the $\{timestamp\}$ of the video, and the following moves are executed: '$\{moves\}$', what will be the arrangement of numbers on the board? Assume that the empty square is represented by '0'. Provide the coordinates of each square along with the corresponding number (e.g., (a,1): 3, (a,2): 0, (b,1): 1, (b,2): 2).

  **Predict Operations**: If the arrangement of numbers on the board is currently in the same state as it was at the $\{timestamp\}$ of the video, what sequence of moves (left, right, up, down) should be executed to achieve the desired number arrangement: '$number\_arrangement$'? Assume that the empty square is represented by '0'. Note that moves cannot push any square beyond the board boundary.

Table 12: Prompt and question templates for the *Number* video demonstration.

**Video**

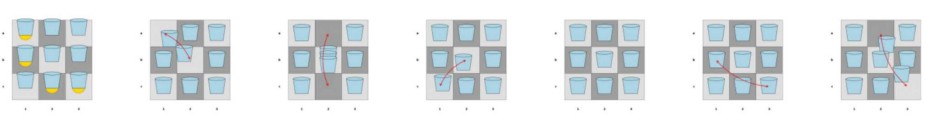

**Full Prompt**
{**task_instruction**}
{**question**}
{**answer_prompt**}

**Task Instruction (Show at Begin)**: The video presents a 'Tricky Cup' puzzle on a $\{N \times N\}$ board. The board is filled with blue cups, each hiding either a yellow coin or nothing underneath. **At the start, all cups are briefly lifted** to reveal what's beneath them. Then, the cups begin a series of moves—each move swaps the positions of two cups, along with their hidden contents. Please carefully watch the video and answer the following question:

**Task Instruction (Show at End)**: The video presents a 'Tricky Cup' puzzle on a $\{N \times N\}$ board. The board is filled with blue cups, each hiding either a yellow coin or nothing underneath. Initially, the contents under the cups are completely hidden. Then, the cups begin a series of moves—each move swaps the positions of two cups, along with their hidden contents. **Toward the end, all cups are briefly lifted** to reveal what's beneath them. Please carefully watch the video and answer the following question:

**Answer Prompt**: Provide a summary of the final answer after 'Final Answer:'

**Questions**
   **Recall Order**: Assume that each time two cups swap their positions, it counts as one move. What are the $\{start\_id\}$ to $\{start\_id\}$ moves shown in the video? For each move, provide the move number and the coordinates of the two cups that swapped positions. Format your response like this: 1st: (a1, b2), 2nd: (c2, b1), 3rd: (a3, c1).

   **Recall Count**: How many times were the cups in the row '$\{row\_idx\}$' involved in the swaps? For each instance, provide the coordinate(s) of the cup(s) before the swap occurred. Format your response like this: 1st: a1, 2nd: a3, 3rd: (a1,a2) (Use a single coordinate for individual cups, or a tuple for multiple cups involved in the same swap.)

   **Infer State**: What are the positions of all the coins at the $\{timestamp\}$ of the video? Provide the coordinates of each coin (e.g., a2, b1, c3).

   **Compare State**: Compare the distribution of contents beneath the cups at the start and end of the video. What are the positions where the contents beneath the cups differ between the two boards? Provide their coordinates along with the corresponding content at the $\{timestamp\}$ of the video. Format your response like this: a1: empty, b3: coin.

   **Predict State**: If the distribution of coins on the board is currently in the same state as it was at the $\{timestamp\}$ of the video, and the following cup swaps are executed in order: '$\{moves\}$', what will be the new distribution of the coins? Provide the coordinates of the coins (e.g., a1, b2).

   **Predict Operations**: If the distribution of coins on the board is currently in the same state as it was at the $\{timestamp\}$ of the video, what sequence of cup swaps should be executed to achieve the desired distribution of coins: 'board'? Format your response as a list of coordinate pairs, such as: (a1, b2), (c3, b1). Each pair represents a single swap between two cups.

Table 13: Prompt and question templates for the *Cup* video demonstration.

**Video**

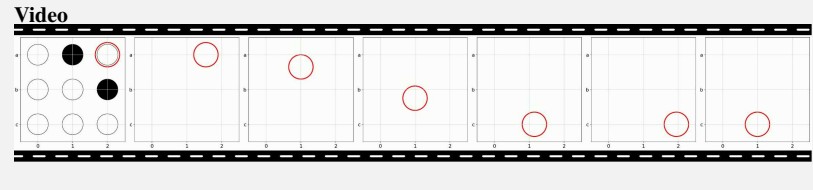

**Full Prompt**
{**task_instruction**}
{**question**}
{**answer_prompt**}

**Task Instruction (Show at Begin)**: The video presents a $\{N \times N\}$ grid. **At the beginning of the video, all positions on the grid are filled with either a black or white piece**. Then, these pieces are visually hidden but still remain in their original positions. A red circle then moves across the grid. Each time the red circle passes by a position on the grid (excluding the starting position), the color of the piece at that position *and* the colors of its immediate orthogonal neighbors (up, down, left, and right) are flipped: black becomes white, and white becomes black. Note that diagonal neighbors are *not* affected. Neighbors are only considered if they exist within the grid's boundaries. Please carefully watch the video and answer the following question:

**Task Instruction (Show at End)**: The video presents a $\{N \times N\}$ grid. All positions on the grid are filled with either a black or white piece. These pieces are visually hidden at the beginning of the video. A red circle then moves across the grid. Each time the red circle passes by a position on the grid (excluding the starting position), the color of the piece at that position *and* the colors of its immediate orthogonal neighbors (up, down, left, and right) are flipped: black becomes white, and white becomes black. Note that diagonal neighbors are *not* affected. Neighbors are only considered if they exist within the grid's boundaries. **The video ends by showing the final arrangement of black and white pieces on the grid**. Please carefully watch the video and answer the following question:

**Answer Prompt**: Provide a summary of the final answer after 'Final Answer:'

**Questions**
   **Recall Order**: Assume that each time the red circle moves from one grid intersection to an adjacent one (horizontally or vertically), it counts as one move. What are the directions (left, right, up, down) of the $\{start\_id\}$ to $\{start\_id\}$ moves made by the red circle in the video? List them in order.
   **Recall Count**: Assume that each time the red circle moves from one grid intersection to an adjacent one (horizontally or vertically), it counts as one move. Given the movement direction '$\{move\}$', how many times does the red circle perform this move? For each occurrence, provide the coordinate of the position before the move (e.g., (a,1), (c,2), etc).
   **Infer State**: What is the arrangement of the black and white pieces on the grid at the $\{timestamp\}$ of the video? Provide each piece's coordinates and color using the format: (column, row): color (e.g., (a,1): black, (c,2): white).
   **Compare State**: Assume that each time the red circle moves from one grid intersection to an adjacent one (horizontally or vertically), it counts as one move. Compare the arrangement of black and white pieces on the grid at the start and end of the video. What are the coordinates where the piece color differ between the two grids? Provide these coordinates along with the corresponding piece color at the $\{timestamp\}$ of the video, using the format: (column, row): color (e.g., (a,1): black, (c,2): white).
   **Predict State**: Assume that each time the red circle moves from one grid intersection to an adjacent one (horizontally or vertically), it counts as one move. If the arrangement of black and white pieces and the position of the red circle on the grid is currently in the same state as it was at the $\{timestamp\}$ of the video, and the following moves are executed: '$\{moves\}$', what will be the arrangement of black and white pieces on the grid? Provide each piece's coordinates and color using the format: (column, row): color (e.g., (a,1): black; (c,2): white).
   **Predict Operations**: Assume that each time the red circle moves from one grid intersection to an adjacent one (horizontally or vertically), it counts as one move. The red circle cannot move beyond the grid boundary. If the arrangement of black and white pieces and the position of the red circle on the grid is currently in the same state as it was at the $\{timestamp\}$ of the video, what sequence of moves (left, right, up, down) should be executed by the red circle to achieve the desired arrangement of black and white pieces: '$\{board\}$'? List them in order.

Table 14: Prompt and question templates for the *Circle* video demonstration.

**Video**

**Full Prompt**
{**task_instruction**}
{**question**}
{**answer_prompt**}

**Task Instruction**: The video demonstrates a series of file manipulation commands executed in the Linux command line. To ensure accurate understanding, note these assumptions:
   * 'touch' commands: All files created by 'touch' do not exist in the target direc try prior to the command's execution.
   * 'rm -r' commands: All files deleted by 'rm -r' do exist in the target directory prior to the command's execution.
   * 'cp' and 'mv' commands: All source files used by 'cp' and 'mv' do exist in the source directory prior to the command's execution.
   * The destination path for 'cp' and 'mv' commands does not contain the target files prior to the command.
Please carefully watch the video and answer the following question:

**Answer Prompt**: Provide a summary of the final answer after 'Final Answer:'

**Questions**
   **Recall Order**: What are the $\{start\_id\}$ to $\{start\_id\}$ $\{cmd\_type\}$commands shown in the video? Provide the order of each command (e.g., 1st, 2nd, 3rd, etc) along with the command content.

   **Recall Count**: How many different $\{file\_type\}$files were involved in the $\{cmd\_type\}$commands throughout the video? Provide the file count along with the specific file names (e.g., 2 '.txt' files: a.txt, b.txt).

   **Infer State**: At the $\{timestamp\}$ of the video, how many $\{file\_type\}$files remain in '$\{path\_name\}$'? Provide the file count along with the specific file names (e.g., 2 '.txt' files: a.txt, b.txt).

   **Compare State**
      - What files were in 'path0/' at the start of the video, but were not there at the end of the video?
      - What files were in 'path0/' at the end of the video, but were not there at the start of the video?
      - After the command '$\{cmd\}$' was executed, what files were in '$\{path\_name1\}$' but were not in '$\{path\_name2\}$'?

   **Predict State**: If the paths currently contain exactly the same files as they did at the $\{timestamp\}$ of the video, and we run the command '$\{cmd\}$', which $\{file\_type\}$files would be in '$\{path\_name\}$'?

   **Predict Operations**: If the paths currently contain exactly the same files as they did at the $\{timestamp\}$ of the video, to ensure that '$\{path\_name\}$' contains exactly the following files: '$\{files\}$', what sequence of commands should be executed? Rules: 1. You may only use the commands 'touch' and 'rm -rf'. 2. You may use at most two commands. 3. Files specified in 'touch' must not appear in 'rm -rf' command, and vice versa (i.e., no overlap). Response Format: If multiple commands are used, separate them with '&'. For example, 'touch path0/{a.txt,b.txt} & rm -rf path0/{c.py,d.json}'.

Table 15: Prompt and question templates for the *File* video demonstration.

**Video**

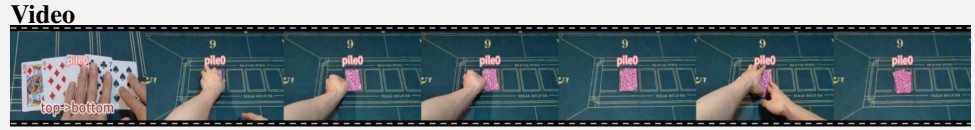

**Full Prompt**
{**task_instruction**}
{**question**}
{**answer_prompt**}

**Task Instruction (Show at Begin)**: The video showcases a sequence of operations involving one or more piles of cards. It **begins by displaying the initial arrangement of cards** in each pile from top to bottom. The cards are then turned face down, after which a series of actions is carried out. Note that there are only two types of actions: adding one card to the top of the pile or removing one card from the bottom of the pile. Please carefully watch the video and answer the following question

**Task Instruction (Show at End)**: The video showcases a sequence of operations involving one or more piles of cards. Throughout the video, before the final reveal of each pile, only two types of actions occur: adding one card to the top of the pile or removing one card from the bottom of the pile. Then, the video **ends by displaying the final arrangement of cards** in each pile from top to bottom. Please carefully watch the video and answer the following question:

**Answer Prompt**: Provide a summary of the final answer after 'Final Answer:'

**Questions**
    **Recall Order**: What are the $\{start\_id\}$ to $\{start\_id\}$ cards being $\{action\_type\}$ any pile throughout the video? For each card, provide the following details: 1. The order (e.g., 1st or 2nd) 2. The suit and value (e.g., 6 of Hearts) 3. The pile involved (e.g., pile0, pile1) Format your response like this: 1st: 6 of Hearts, pile0 2nd: Jack of Spades, pile1.
    **Recall Count**: How many cards were $\{action\_type\}$ '$\{pile\_name\}$' throughout the video? For each card, provide its suit and value (e.g., 6 of Hearts) Format your response like this: 2 cards: 6 of Hearts, King of Clubs.
    **Infer State**: At the $\{timestamp\}$ of the video, what cards are in '$\{pile\_name\}$'? List them in order from top to bottom, including both the value and suit of each card. Format your response like this: 6 of Hearts, King of Clubs, 3 of Spades.
    **Compare State**: What cards were in '$\{pile\_name\}$' at the $\{timestamp\}$ of the video, but were not there at the $\{timestamp2\}$ of the video? For each card, provide its suit and value. Format your response like this: 6 of Hearts, King of Clubs, 3 of Spades.
    **Predict State**: If the piles currently contain exactly the same cards as they did at the $\{timestamp\}$ of the video, and now we perform these actions in order: '$\{actions\}$'. What cards would be in '$\{pile\_name\}$'? List them in order from top to bottom, including both the value and suit of each card. Format your response like this: 6 of Hearts, King of Clubs, 3 of Spades.
    **Predict Operations**: If the piles currently contain exactly the same cards as they did at the $\{timestamp\}$ of the video, to ensure that '$\{pile\_name\}$' contains exactly the following cards from top to bottom: '$\{cards\}$', what sequence of actions should be performed? Rules: 1. Each action must either add a card to a pile or remove a card from a pile. 2. You may only add cards to the top of a pile or remove cards from the bottom of a pile. Response Format: List the actions in sequence, specifying the action, card, and pile. Separate each action with a comma. For example, 'add 6 of Hearts to pile0, remove King of Clubs from pile0'

Table 16: Prompt and question templates for the *Card* video demonstration.

**Video**

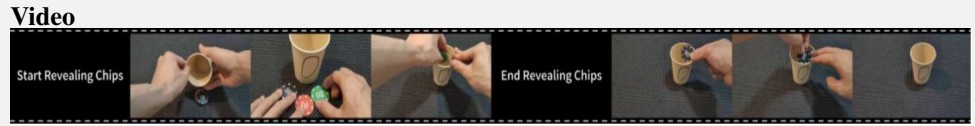

**Full Prompt**
{**task_instruction**}
{**question**}
{**answer_prompt**}

**Task Instruction (Show at Begin)**: The video showcases a sequence of operations involving one or more cup(s) and chips. It **begins by showing the initial chips** contained in each cup. Then, a series of actions are carried out. Note that there are only two types of actions: adding one chip to a cup or removing one chip from a cup. Please carefully watch the video and answer the following question:

**Task Instruction (Show at End)**: The video showcases a sequence of operations involving one or more cup(s) and chips. Throughout the video, before the final reveal of chips contained in each cup, only two types of actions occur: adding one chip to a cup or removing one chip from a cup. Then, the video **ends by displaying the final chips** contained in each cup. Please carefully watch the video and answer the following question:

**Answer Prompt**: Provide a summary of the final answer after 'Final Answer:'

**Questions**
   **Recall Order**: Disregarding the process of revealing all chips in the cup(s), what are the {$start\_id$} to {$start\_id$} chips being {$action\_type$} any cup throughout the video? For each chip, provide the following details: 1. The order (e.g., 1st or 2nd) 2. The value (e.g., 20) 3. The cup involved (e.g., cup0, cup1) Format your response like this: 1st: 100, cup0 2nd: 20, cup1.
   **Recall Count**: Disregarding the process of revealing all chips in the cup(s), how many chips were {$action\_type$} '{$cup\_name$}' throughout the video? For each chip, provide its value (order does not matter). Format your response like this: 4 chips: 20, 5, 100, 100.
   **Infer State**: At the {$timestamp$} of the video, how many chips were in '{$cup\_name$}'? For each chip, provide its value (order does not matter). Format your response like this: 4 chips: 20, 5, 100, 100.
   **Compare State**: At which point in the video is the total value of chips in '{$cup\_name$}' higher, at {$timestamp1$} or {$timestamp2$}? Also, what is the difference in value between the two times? Format your response like this: "time_with_higher_value", "difference_in_value" (e.g., start, 115).
   **Predict State**: If the cups currently contain exactly the same chips as they did at the {$timestamp$} of the video, and now we perform these actions in order: '{$actions$}'. How many chips would be in '{$cup\_name$}'? For each chip, provide its value (order does not matter). Format your response like this: 4 chips: 20, 5, 100, 100.
   **Predict Operations**: If the cups currently contain exactly the same chips as they did at the {$timestamp$} of the video, to ensure that '{$cup\_name$}' contains exactly the following chips: '{$chips$}' (order does not matter), what sequence of actions should be performed? Rules: 1. Each action must either add a chip to a cup or remove a chip from a cup. 2. Available chips for addition are: 5, 10, 20, 50, 100. 3. You may only remove a chip if it is already present in the cup. Response Format: List the actions in sequence, specifying the action, chip, and cup. Separate each action with a comma. For example, 'add 20 to cup0, remove 50 cup0'

Table 17: Prompt and question templates for the *Chip* video demonstration.

You will be given a question, a model response and a ground-truth answer. Your task is to determine whether the model response is correct based on the ground-truth answer. The model response should contain all information in the ground-truth answer.
Question: {*question*}
Model Response: {*response*}
Ground-Truth Answer: {*ground_truth*}
Directly output "Correct" or "Incorrect":

Table 18: Prompt used for LLM-based evaluation.

**Number & Circle**: You will be given a model-generated response describing a sequence of movements. Your task is to extract the movements in the order they appear and return them as a list (e.g., ['left', 'up', 'down', 'right']).
Model Response: {*response*}
Extracted Movements:

**Cup**: You will be given a model-generated response describing a sequence of cup swaps. Each swap is represented as a pair of coordinates—for example, (a1, b2)—indicating the two positions being swapped.
Your task: Extract all coordinate pairs from the response in the exact order they appear, and return them as a list of tuples.
Format your answer like this: [('a1', 'b2'), ('c1', 'b1'), ('a3', 'b2')]
Model Response: {*response*}
Extracted Swaps:

**File**: You will be given a model-generated response regarding a file operation command in Linux system.
Your task: Identify and extract only the actual command from the model response, removing any irrelevant or descriptive text.
Model Response: {*response*}
Extracted Command:

**Card**: You will be given a model-generated response describing a sequence of operations performed to cards. Each operation either adds or removes a card from pile0 or pile1.
Your task:
- Extract all valid operations and return them as a list of strings.
- Each operation must involve either adding or removing a card to or from pile0 or pile1.
- If no valid operations are found, return an empty list ([]).
Format your answer like this: ['add 6 of Hearts to pile0', 'remove King of Clubs from pile0']
Model Response: {*response*}
Extracted Operations:

**Chip**: You will be given a model-generated response describing a sequence of operations involving chips and cups. Each operation either adds or removes a chip from cup0 or cup1.
Your task:
- Extract all valid operations and return them as a list of strings.
- Each operation must involve either adding or removing a chip to or from cup0 or cup1.
- If no valid operations are found, return an empty list ([]).
Format your answer like this: ['add 20 to cup0', 'remove 50 cup0']
Model Response: {*response*}
Extracted Operations:

Table 19: Prompts used for operation extraction.

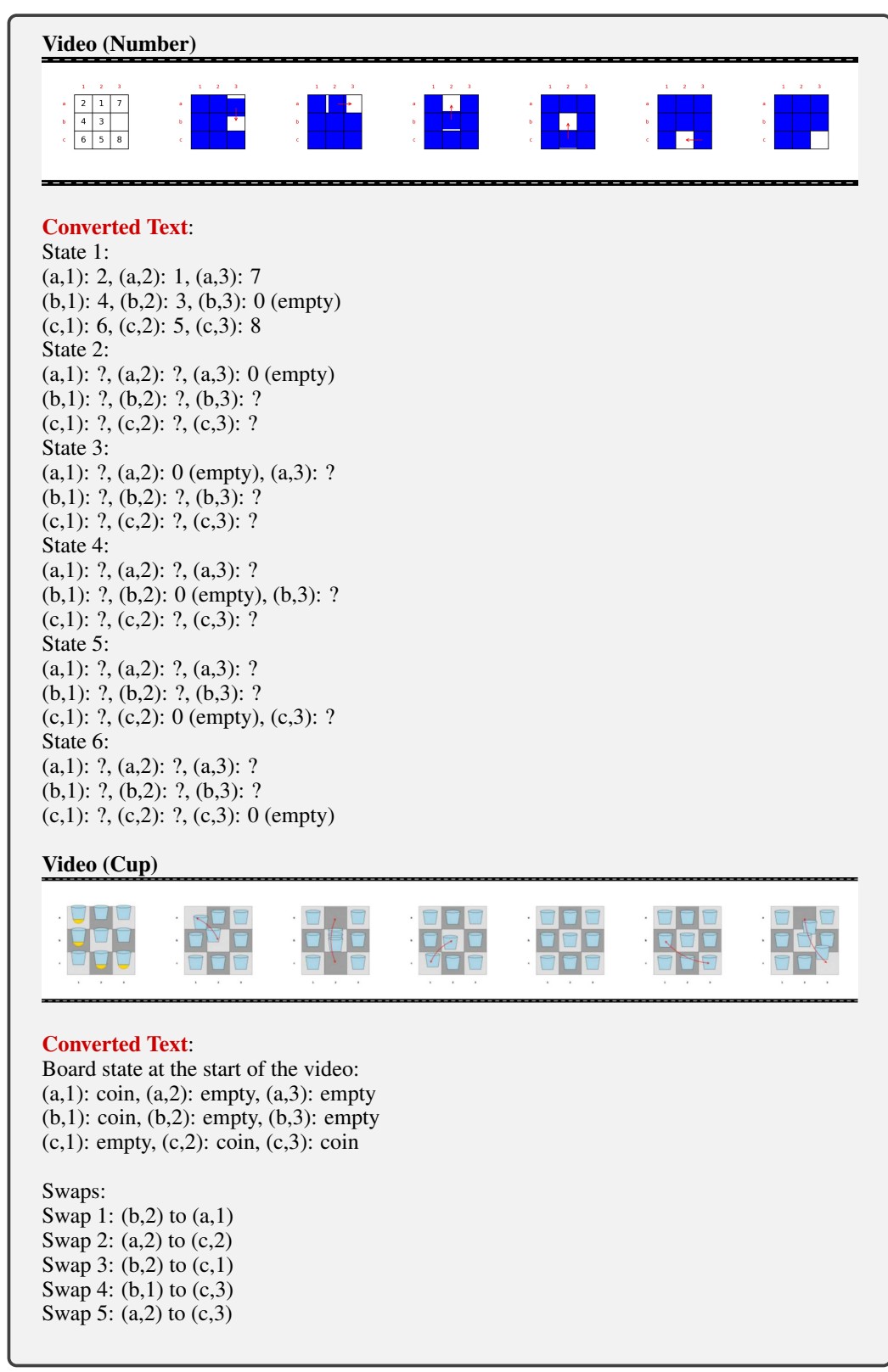

**Video (Number)**

**Converted Text**:
State 1:
(a,1): 2, (a,2): 1, (a,3): 7
(b,1): 4, (b,2): 3, (b,3): 0 (empty)
(c,1): 6, (c,2): 5, (c,3): 8
State 2:
(a,1): ?, (a,2): ?, (a,3): 0 (empty)
(b,1): ?, (b,2): ?, (b,3): ?
(c,1): ?, (c,2): ?, (c,3): ?
State 3:
(a,1): ?, (a,2): 0 (empty), (a,3): ?
(b,1): ?, (b,2): ?, (b,3): ?
(c,1): ?, (c,2): ?, (c,3): ?
State 4:
(a,1): ?, (a,2): ?, (a,3): ?
(b,1): ?, (b,2): 0 (empty), (b,3): ?
(c,1): ?, (c,2): ?, (c,3): ?
State 5:
(a,1): ?, (a,2): ?, (a,3): ?
(b,1): ?, (b,2): ?, (b,3): ?
(c,1): ?, (c,2): 0 (empty), (c,3): ?
State 6:
(a,1): ?, (a,2): ?, (a,3): ?
(b,1): ?, (b,2): ?, (b,3): ?
(c,1): ?, (c,2): ?, (c,3): 0 (empty)

**Video (Cup)**

**Converted Text**:
Board state at the start of the video:
(a,1): coin, (a,2): empty, (a,3): empty
(b,1): coin, (b,2): empty, (b,3): empty
(c,1): empty, (c,2): coin, (c,3): coin

Swaps:
Swap 1: (b,2) to (a,1)
Swap 2: (a,2) to (c,2)
Swap 3: (b,2) to (c,1)
Swap 4: (b,1) to (c,3)
Swap 5: (a,2) to (c,3)

Table 20: Illustration of how to convert key video information into text ("vid2txt") for *Number* and *Cup* videos.

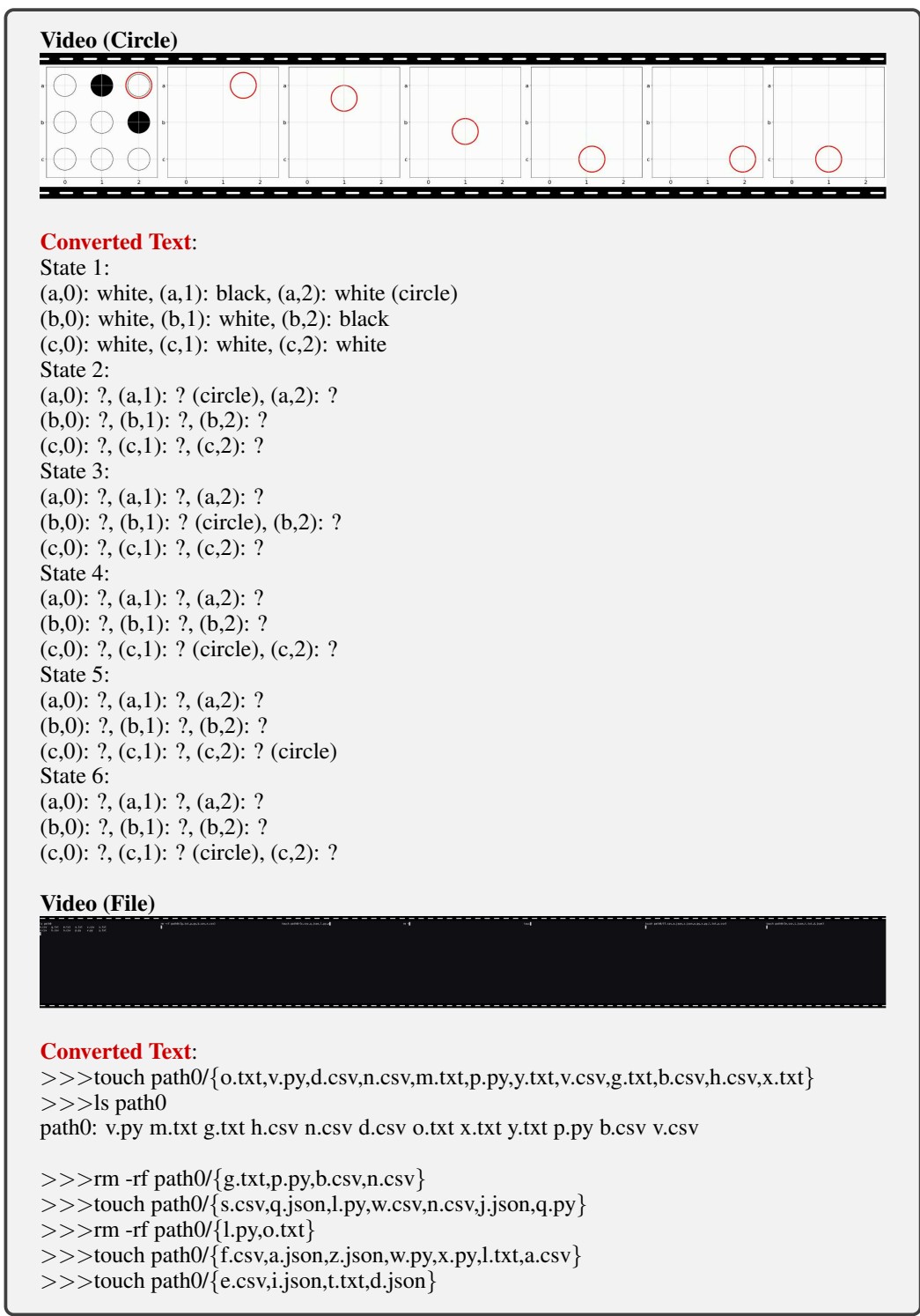

Table 21: Illustration of how to convert key video information into text ("vid2txt") for *Circle* and *File* videos.

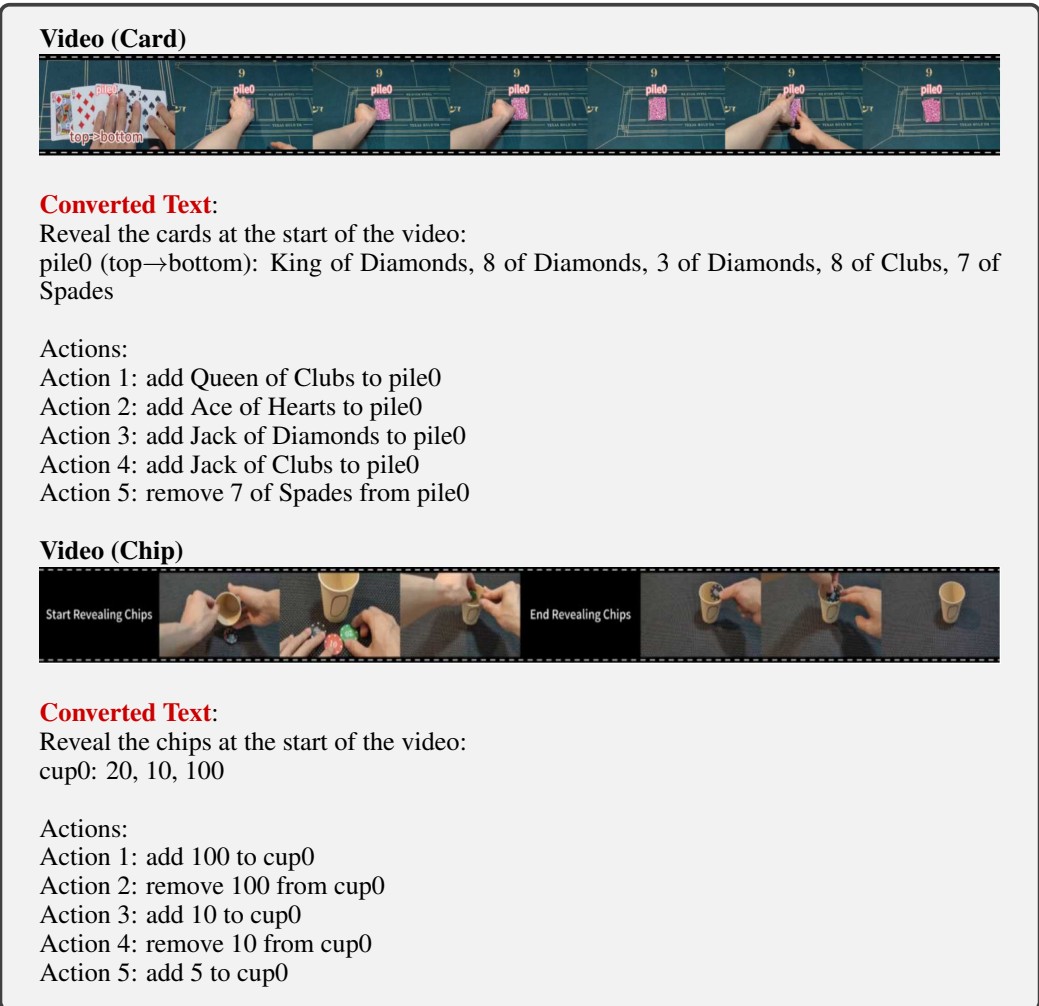

**Video (Card)**

**Converted Text**:
Reveal the cards at the start of the video:
pile0 (top→bottom): King of Diamonds, 8 of Diamonds, 3 of Diamonds, 8 of Clubs, 7 of Spades

Actions:
Action 1: add Queen of Clubs to pile0
Action 2: add Ace of Hearts to pile0
Action 3: add Jack of Diamonds to pile0
Action 4: add Jack of Clubs to pile0
Action 5: remove 7 of Spades from pile0

**Video (Chip)**

**Converted Text**:
Reveal the chips at the start of the video:
cup0: 20, 10, 100

Actions:
Action 1: add 100 to cup0
Action 2: remove 100 from cup0
Action 3: add 10 to cup0
Action 4: remove 10 from cup0
Action 5: add 5 to cup0

Table 22: Illustration of how to convert key video information into text ("vid2txt") for *Card* and *Chip* videos.

Table 23: Inference configurations for the evaluated MLLMs. "1fps/$N$" indicates that videos with a duration $\leq N$ seconds are processed at 1fps, while for videos longer than $N$ seconds, $N$ frames are uniformly sampled. *The temperature is set to 0.0 by default and increased to 1.0 if the response exceeds the "Max New Tokens" limit—this adjustment is applied in token count experiments to prevent excessively long responses with repeated tokens.

| Model | Version | Input Frames | Temperature | Max New Tokens |
|---|---|---|---|---|
| **Proprietary Models** | | | | |
| GPT-4o | `gpt-4o-2024-11-20` | 1fps/50 | 1.0 | 8,192 |
| o4-mini | `o4-mini-2025-04-16` | 1fps/50 | 1.0 | 8,192 |
| Seed1.5-VL | `doubao-1-5-thinking-vision-pro-250428` | 1fps/128 | 0.0 | 16,384 |
| Gemini-2.0-Flash | `gemini-2.0-flash` | 1fps | 0.0 | 4,096 |
| Gemini-2.5-Flash | `gemini-2.5-flash-preview-04-17` | 1fps | 0.0/1.0* | 65,536 |
| Gemini-2.5-Pro | `gemini-2.5-pro-preview-05-06` | 1fps | 0.0/1.0* | 65,536 |
| **Open-source Models** | | | | |
| *Efficient Models* | | | | |
| mPLUG-Owl3 | `mPLUG-Owl3-7B-240728` | 32 | 1.0 | 1,024 |
| MiniCPM-V 2.6 | `MiniCPM-V-2_6` | 32 | 1.0 | 1,024 |
| MiniCPM-o 2.6 | `MiniCPM-o-2_6` | 32 | 1.0 | 1,024 |
| LLaVA-OneVision-7B | `llava-onevision-qwen2-7b-ov` | 32 | 0.0 | 2,048 |
| LLaVA-Video-7B | `LLaVA-Video-7B-Qwen2` | 32 | 0.0 | 2,048 |
| InternVL3-8B | `InternVL3-8B` | 32 | 0.0 | 4,096 |
| Qwen2.5-VL-7B | `Qwen2.5-VL-7B-Instruct` | 32 | 0.01 | 4,096 |
| Kimi-VL-A3B | `Kimi-VL-A3B-Instruct` | 32 | 1.0 | 16,384 |
| *Flagship Models* | | | | |
| LLaVA-OneVision-72B | `llava-onevision-qwen2-72b-ov-sft` | 64 | 0.0 | 2,048 |
| LLaVA-Video-72B | `LLaVA-Video-72B-Qwen2` | 64 | 0.0 | 2,048 |
| InternVL3-78B | `InternVL3-78B` | 64 | 0.0 | 4,096 |
| Qwen2.5-VL-72B | `Qwen2.5-VL-72B-Instruct` | 64 | 0.01 | 4,096 |

