# OpenReview forum: "VideoReasonBench: Can MLLMs Perform Vision-Centric Complex Video Reasoning?"
_ICLR.cc/2026/Conference — ICLR 2026 Poster_

### Official Review · Reviewer_4Dkk · 2025-10-30

**Soundness:** 3
**Presentation:** 3
**Contribution:** 3
**Rating:** 6
**Confidence:** 4

**Summary:**

This paper proposed a video question answering benchmark focusing on the visual reasoning ability of the MLLMs. In specifice, the videos are evenly created for six tasks and the question requires the compherehensive understanding to answer. Experiemnts shows that existing MLLMs struggles in answering the questions, and enableing thinking ability significantly boosts the models' performance.

**Strengths:**

● The paper is well motivated. Evaluating the visual reasoning ability of MLLMs can be crutial.
● The proposed benchmark is of reasonable size and distribution.
● Experiments verified the proposed benchmark is challanging for existing MLLMs, and performance gain of CoT supports the claim that the questions require visual reasoning to solve.

**Weaknesses:**

● The proopsed data construction pipeline is highly specialized for the proposed types of questions, and can hardly generalize to boarder question types.
● In table 2, the experiment shows that vid2txt setting significantly outperforms traditional VidQA setting (over 48% on gemini-2.5-flash). This somewhat violates the claim that the benchmark is challanging the visual centric reasoning ability of MLLMs as this implies that the perceptual ability seems playing a more important role  than reasoning for the low performance of current MLLMs.

**Questions:**

● The comparison of vid2txt setting in table 2 seems suggensting that failure of current MLLMs are mainly caused by the percetion failure rather than reasoning failure as the performance gap between vid2txt setting and traditional VidQA setting is significantly larger than that between models w./w.o. thinking. Therefore, the main challenge of this benchmark is perception rather than reasoning. How do author explain this performance gap and its contradictory with the main claim?

---

> ### Author Response · Authors · 2025-11-25
> **Response to Reviewer 4Dkk (1/1)**
>
> Dear Reviewer 4Dkk,
>
> Thank you for the thoughtful review of our work. We sincerely appreciate your positive assessment, including your comments that the work is `well motivated`, that our benchmark has a `reasonable size and distribution`, and that it is `challenging and indeed requires visual reasoning to solve`. Below, we provide point-by-point responses to your questions and concerns.
>
> >W1: The proopsed data construction pipeline is highly specialized for the proposed types of questions, and can hardly generalize to boarder question types.
>
> We thank the reviewer for this observation. We acknowledge that our data construction pipeline is tailored to the specific tasks in **VideoReasonBench**. However, we would like to clarify that this specialization is not a limitation of our work, but rather an intentional design choice to ensure data quality. We address this concern from three perspectives:
>
> **1. Clarifying the intended contribution and scope**
>
> While developing a general-purpose generative pipeline for video reasoning data is valuable, it is beyond the scope of this work. Our primary contribution is VideoReasonBench itself: a diagnostic tool designed to expose the specific deficits of current MLLMs in vision-centric complex video reasoning. The pipeline is designed to support this goal by automatically and accuractely producing the questions and answers.
>
> **2. Specialized pipeline ensures zero-noise ground truth**
>
> Most existing benchmarks often rely on LLM-generated or crowd-sourced annotations. While these approaches are task-agnostic, they are prone to hallucinations, ambiguity, and errors. In contrast, our specialized programmatic pipeline derives both questions and answers directly from ground-truth state transition sequences. This ensures fully deterministic, error-free annotation.
>
> **3. The underlying framework is generalizable**
>
> While the specific data generation pipeline is tailored to our six tasks, the underlying principled framework—modeling videos as a sequence of State Transitions—is generalizable. It applies to a broad range of real-world applications such as:
> - **GUI navigation**: Tracking and predicting system states based on observed on-screen actions.
> - **Robotics and Embodied AI**: Planning manipulation sequences by understanding history action and state transision sequences.
> - **Surveillance Analysis**: Tracking the existence of certain subjects in an environment.
>
> >W2: The comparison of vid2txt setting in table 2 seems suggensting that failure of current MLLMs are mainly caused by the percetion failure rather than reasoning failure. How do author explain this performance gap and its contradictory with the main claim?
>
> Thank you for the thoughtful question. While we agree that perception is a key bottleneck for current MLLMs, we respectfully disagree that the main challenge of our benchmark is perception rather than reasoning.
>
> The most critical evidence lies in the performance within the `vid2txt` setting itself (the table below). Even when provided with the "perfect" text representation (removing the perception barrier), non-thinking models still perform very poorly. For example, the non-thinking Gemini-2.0-Flash lags behind the two proprietary thinking models by almost 30% accuracy. Similarly, non-thinking open-source models such as Qwen2.5-VL-7B/72B fall short by nearly 50%. These gaps are on the same scale as the gap between vid2txt and traditional VidQA setting.
>
> Furthermore, it is worth noting that even the thinking-enhanced models still struggle under the `vid2txt` setting, especially on Level 3 Predict problems.
>
> Taken together, these findings show that while perception is indeed challenging, complex reasoning remains a substantial and independent difficulty. Thus, the observed gap is not contradictory to our main claim; rather, it highlights that our benchmark stresses both dimensions of vision-centric complex video reasoning.
>
> **Results under `vid2txt` setting**
> | Model             | Think | Recall Order | Recall Count | Infer State | Compare State | Predict State | Predict Operation | Avg  |
> |-------------------|-------|--------------|--------------|-------------|----------------|----------------|--------------------|------|
> | Qwen2.5-VL-7B     | ✘     | 32.1         | 30.8         | 7.9         | 13.3           | 1.7            | 1.7                | 14.5 |
> | Qwen2.5-VL-72B    | ✘     | 62.5         | 50.0         | 7.5         | 7.5            | 1.2            | 5.0                | 22.3 |
> | Gemini-2.0-Flash  | ✘     | 66.7         | 52.5         | 42.9        | 37.1           | 26.2           | 20.0               | 40.9 |
> | Seed1.5-VL        | ✔     | 83.3         | 87.9         | 74.2        | 71.7           | 54.2           | 45.4               | 69.4 |
> | Gemini-2.5-Flash  | ✔     | 86.7         | 82.5         | 84.2        | 75.8           | 56.2           | 47.9               | 72.2 |

---

> > ### Comment · Reviewer_4Dkk · 2025-11-26
> >
> > Thanks for the response, I'd like to keep my rating unchanged.

---

> > > ### Author Response · Authors · 2025-11-28
> > >
> > > Thank you for the prompt reply and for maintaining the positive rating. If there are any remaining points we can further clarify, please feel free to let us know. Thank you again for your valuable comments!

---

### Official Review · Reviewer_qkUM · 2025-11-01

**Soundness:** 3
**Presentation:** 3
**Contribution:** 3
**Rating:** 6
**Confidence:** 3

**Summary:**

This paper introduces a diagnostic benchmark for video reasoning that treats each clip as a sequence of visible operations acting on a partially hidden state. Tasks span perception, latent-state inference, and counterfactual prediction across both synthetic and real scenes with programmatically generated ground truth. Evaluations show current multimodal models struggle, especially when fine-grained temporal perception and long-horizon reasoning are required; converting videos to textual state–operation traces and allowing longer chains of thought markedly narrows the gap. The paper highlights perception bottlenecks, the value of explicit reasoning, and the need for methods that integrate stronger visual tracking, external memory, and executable reasoning.

**Strengths:**

1. The benchmark formalizes video as “latent state + a sequence of visible operations,” decomposed into three reasoning levels with six concrete skills, which makes failure modes inspectable and comparable.

2. The study evaluates a wide span of open-source MLLMs

**Weaknesses:**

1. Since most tasks are scored with a judge LLM, would you consider reporting a brief robustness study (e.g., prompt ablations, temperature sweeps, and an alternative judge model) and an agreement statistic?

2. To make cross-model comparisons easier, could you add some description about budget setting, such as fps/frame count and max tokens?

3. The results indicate accuracy drops as state size and operation count increase, and when the final state is only revealed at the end. What may happen if we extend scenarios to minutes-scale durations or larger state spaces?

**Questions:**

Please refer to the weakness

---

> ### Author Response · Authors · 2025-11-25
> **Response to Reviewer qkUM (1/2)**
>
> Dear Reviewer qkUM,
>
> We greatly appreciate your thoughtful feedback on our work! We are encouraged by your recognition of our benchmark design, which makes `failure modes inspectable and comparable`. We address your questions and concerns as follows:
>
> >W1: Since most tasks are scored with a judge LLM, would you consider reporting a brief robustness study (e.g., prompt ablations, temperature sweeps, and an alternative judge model) and an agreement statistic?
>
> We thank the reviewer for this constructive suggestion. In response, we conducted additional analyses to quantify the robustness of our evaluation. Specifically, we (1) employ an alternative judge model (Qwen3-235B-A22B) and compare its assessments with those produced by our original judge model, Qwen2.5-72B; and (2) paraphrase the MLLMs' answers and compare the resulting evaluations with those based on the original answers.
>
> As shown in the tables below, the results remain highly consistent across both analyses: average accuracy differences stay within 1%, and the agreement rate exceeds 98.7%. These findings indicate that the evaluation is stable with respect to both answer reformulation and judge model choice.
>
> We also manually reviewed 200 randomly sampled responses from Gemini‑2.5‑Pro, Gemini‑2.0‑Flash, Seed1.5‑VL and Qwen2.5-VL-7B (50 samples from each model). In every case, Qwen2.5‑72B’s judgement matched our own assessments, demonstrating the corectness of evaluation.
>
> **Results with Different Answers** ( Inter-Answer Form Agreement=98.7%)
> | Model                    | Answer Form   | Recall Order | Recall Count | Infer State | Compare State | Predict State | Avg   |
> |--------------------------|---------------|--------------|--------------|-------------|----------------|----------------|-------|
> | Gemini-2.5-Flash         | original      | 44.6         | 41.7         | 27.9        | 27.1           | 13.8           | 31.0  |
> | Gemini-2.5-Flash         | paraphrased   | 45.4         | 44.2         | 28.8        | 27.1           | 13.8           | 31.8  |
> | Gemini-2.5-Flash w/o think | original    | 22.5         | 34.2         | 19.6        | 20.4           | 8.8            | 21.1  |
> | Gemini-2.5-Flash w/o think | paraphrased | 22.3         | 34.5         | 20.8        | 19.5           | 10.2           | 21.2  |
> | Seed1.5-VL               | original      | 24.2         | 27.1         | 3.8         | 7.9            | 3.8            | 13.33 |
> | Seed1.5-VL               | paraphrased   | 24.2         | 26.7         | 3.8         | 7.9            | 3.8            | 13.25 |
>
> **Results with Different Judge Models**  (Inter-Judge Agreement=98.9%)
> | Model                   | Judge Model        | Recall Order | Recall Count | Infer State | Compare State | Predict State | Avg  |
> |-------------------------|--------------------|--------------|--------------|-------------|----------------|---------------|------|
> | Gemini-2.5-Flash        | Qwen2.5-72B        | 44.6         | 41.7         | 27.9        | 27.1           | 13.8          | 31.0 |
> | Gemini-2.5-Flash        | Qwen3-235B-A22B    | 45.8         | 40.8         | 27.1        | 26.2           | 14.2          | 30.8 |
> | Gemini-2.5-Flash w/o think | Qwen2.5-72B     | 22.5         | 34.2         | 19.6        | 20.4           | 8.8           | 21.1 |
> | Gemini-2.5-Flash w/o think | Qwen3-235B-A22B | 22.5         | 32.9         | 19.6        | 20.4           | 9.6           | 21.0 |
> | Seed1.5-VL              | Qwen2.5-72B        | 24.2         | 27.1         | 3.8         | 7.9            | 3.8           | 13.3 |
> | Seed1.5-VL              | Qwen3-235B-A22B    | 23.8         | 25.8         | 2.1         | 6.7            | 3.8           | 12.4 |
>
> > W2: To make cross-model comparisons easier, could you add some description about budget setting, such as fps/frame count and max tokens?
>
> Thank you for the helpful suggestion. Relevant details are documented in Appendix C.1 and Table 18, and we have now clarified the reference in the main text (Section 3.1). Specifically, proprietary models with longer context windows process videos at 1 fps with an upper limit on the total number of frames. For open-source models, we instead apply uniform frame sampling with fixed budgets: 32 frames for efficient models and 64 frames for flagship models. The `max tokens` setting follows each model’s default official inference configuration whenever available; if no standard setting exists, we select a token budget consistent with the model’s demonstrated capacity for chain-of-thought reasoning.

---

> > ### Author Response · Authors · 2025-11-25
> > **Response to Reviewer qkUM (2/2)**
> >
> > >W3: The results indicate accuracy drops as state size and operation count increase, and when the final state is only revealed at the end. What may happen if we extend scenarios to minutes-scale durations or larger state spaces?
> >
> > Interesting question! To further investigate scalability, we extended both the state size and the operation count in the *Number* and *Cup* demonstrations using our automatic video and QA generation engine. As shown in the following results, model performance decreases as either dimension grows. Notably, however, Gemini-2.5-Pro exhibits a smaller relative drop than Gemini-2.5-Flash and Seed1.5-VL, suggesting that it is more resilient to increases in video complexity.
> >
> > We also view state size and operation count scaling as practical strategies for increasing benchmark difficulty in future iterations. We appreciate the reviewer’s insightful comment and believe the expanded experiments meaningfully strengthen the paper.
> >
> > **Results of Extending Board Size, num_op=5-14**
> > | Model             | 3x3 | 4x4 | 5x5 | 6x6 |
> > |-------------------|-----|-----|-----|-----|
> > | Seed1.5-VL        | 5.4 | 6.7 | 5.8 | 3.3 |
> > | Gemini-2.5-Flash  | 25.8| 19.2| 7.9 | 6.7 |
> > | Gemini-2.5-Pro    | 70.4| 61.7| 59.6| 56.7 |
> >
> > **Results of Extending Operation Count, board_size=3x3,4x4**
> > | Model           | 5-9 (20.0s)  | 10-14 (32.4s) | 15-19 (44.7s) | 20-24 (57.1s) | 25-29 (69.4s) |
> > |-----------------|------|-------|-------|-------|-------|
> > | Seed1.5-VL      | 10.4 | 1.7   | 0.4   | 0.4   | 0.0   |
> > | Gemini-2.5-Flash| 25.8 | 19.2  | 9.6   | 5.8   | 5.0   |
> > | Gemini-2.5-Pro  | 68.8 | 63.3  | 61.2  | 55.4  | 51.7  |

---

> > > ### Author Response · Authors · 2025-11-28
> > > **Looking Forward to Your Feedback**
> > >
> > > Dear Reviewer qkUM,
> > >
> > > We sincerely appreciate the time and effort you have invested in reviewing our paper. As the discussion period concludes on December 3rd, we would like to gently check whether our responses have sufficiently addressed your concerns. If any questions remain or if there are points that would benefit from additional clarification, we would be more than happy to provide further details. On the other hand, if our rebuttal has resolved your concerns, we would be truly grateful if this could be reflected in your final evaluation.
> > >
> > > Thank you again for your thoughtful review. We look forward to your feedback.
> > >
> > > Sincerely,
> > >
> > > VideoReasonBench Authors

---

### Official Review · Reviewer_tQvt · 2025-11-01

**Soundness:** 4
**Presentation:** 2
**Contribution:** 2
**Rating:** 4
**Confidence:** 4

**Summary:**

The paper presents VideoReasonBench, a vision-focused benchmark that emphasizes step-wise reasoning over long sequences of visually observable operations on partially observable latent states such as sliding-tile board states. It defines three increasing skill levels: recall for fine-grained operation perception, infer for latent state inference, and predict for counterfactual planning. It includes six demonstration types: Number, Circle, Cup, File, Card, and Chip. The dataset contains 240 videos and 1,440 questions, with structured prompts and automatic answer generation based on state-transition scripts. Evaluation uses a text-only LLM judge; for Predict Operation, it simulates model-proposed operations to verify target states. Among 18 MLLMs, most models perform poorly with less than 10 percent accuracy overall. Gemini-2.5-Pro in thinking mode reaches 56 percent, and the benchmark shows strong sensitivity to thinking budget and heavy reliance on vision compared to earlier video benchmarks.

**Strengths:**

The benchmark covers both synthetic and real-world capture paths, including Matplotlib programmatic renderings, terminal screenshots, and manual videos, with balanced distributions across skills and demos and controlled settings for state size and operation length to adjust difficulty.

**Weaknesses:**

W1: For five of six skills, correctness is determined by a text-only LLM judge given GT and the model output. Even with careful prompts, LLM graders can show bias or instability and may prefer format-matched answers over truly correct content. The paper does not report inter-judge agreement, self-consistency, or adversarial sensitivity such as paraphrase, verbosity, or distractors. This is quite concerning.
For Predict Operation, answers are extracted using an LLM, then simulated. This introduces another potential failure point in extraction, but there is no error analysis distinguishing extraction failures from planning failures.
vid2txt is generated from the ground-truth state transitions. It works well for isolating perception versus reasoning, but it effectively provides a clean event log the models do not need to perceive. It is possible that the performance gap partly reflects a best-case text abstraction that few real video systems could realistically obtain.


W2: The authors stated that the videos and questions in these benchmarks often resemble those in earlier benchmarks and fall short in demanding deeper video reasoning. It is unclear how this dataset helps address that issue. It is also unclear in what way the previous ones resemble.



W3: According to the paper, all videos are synthetic or semi-synthetic demonstrations built from controlled state-transition scripts rather than natural videos.
It seems the dataset favors discrete and symbolic reasoning. All operations appear atomic and reversible, meaning the benchmark strongly favors models that can internally represent counting, position updates, or ordering. This raises the concern that it may overfit to symbolic reasoning rather than perceptual or intuitive physics reasoning.
It may also underestimate challenges such as motion blur, occlusion, object continuity, or ambiguous visual events found in open-world settings. But this is a minor concern.

**Questions:**

How consistent is the LLM-as-judge across paraphrased or differently formatted answers?

Can you analyze the potential bias introduced by the six procedural video types and clarify whether these discrete settings generalize to naturalistic visual reasoning?

---

> ### Author Response · Authors · 2025-11-25
> **Response to Reviewer tQvt (1/5)**
>
> Dear Reviewer tQvt,
>
> Thank you very much for your careful review and thoughtful comments on our work! Please find our responses to your questions and concerns below.
>
> >W1: For five of six skills, correctness is determined by a text-only LLM judge given GT and the model output. Even with careful prompts, LLM graders can show bias or instability and may prefer format-matched answers over truly correct content. The paper does not report inter-judge agreement, self-consistency, or adversarial sensitivity such as paraphrase, verbosity, or distractors. How consistent is the LLM-as-judge across paraphrased or differently formatted answers?
>
> Thank you for highlighting this important consideration. We fully agree that the reliability and stability of an LLM-based judge is crucial. To mitigate format-related bias and improve stability, we carefully design task instructions (Tables 7–12 in the appendix), prompting MLLMs to explicitly summarize their final answer in specific formats. This allows the judge model (Qwen2.5-72B) to perform a direct comparison between format-similar model predictions and ground-truth answers—an ability well within its demonstrated strengths.
>
> In light of your constructive suggestion, we conducted additional analyses to quantify the robustness of our evaluation. Specifically, we (1) paraphrase the MLLMs' answers and compare the evaluation results to that of the original answers, (2) utilize a different judge model (Qwen3-235B-A22B) and compare its evaluations with those of Qwen2.5-72B.
>
> As shown in the tables below, the results remain highly consistent across both analyses: average accuracy differences stay within 1%, and the agreement rate exceeds 98.7%. These findings indicate that the evaluation is stable with respect to both answer reformulation and judge model choice.
>
> We also manually reviewed 200 randomly sampled responses from Gemini‑2.5‑Pro, Gemini‑2.0‑Flash, Seed1.5‑VL and Qwen2.5-VL-7B (50 samples from each model). In every case, Qwen2.5‑72B’s judgement matched our own assessments, demonstrating the corectness of evaluation.
>
> **Results with Different Answers** (Inter-Answer Form Agreement=98.7%)
> | Model                    | Answer Form   | Recall Order | Recall Count | Infer State | Compare State | Predict State | Avg   |
> |--------------------------|---------------|--------------|--------------|-------------|----------------|----------------|-------|
> | Gemini-2.5-Flash         | original      | 44.6         | 41.7         | 27.9        | 27.1           | 13.8           | 31.0  |
> | Gemini-2.5-Flash         | paraphrased   | 45.4         | 44.2         | 28.8        | 27.1           | 13.8           | 31.8  |
> | Gemini-2.5-Flash w/o think | original    | 22.5         | 34.2         | 19.6        | 20.4           | 8.8            | 21.1  |
> | Gemini-2.5-Flash w/o think | paraphrased | 22.3         | 34.5         | 20.8        | 19.5           | 10.2           | 21.2  |
> | Seed1.5-VL               | original      | 24.2         | 27.1         | 3.8         | 7.9            | 3.8            | 13.33 |
> | Seed1.5-VL               | paraphrased   | 24.2         | 26.7         | 3.8         | 7.9            | 3.8            | 13.25 |
>
> **Results with Different Judge Models**  (Inter-Judge Agreement=98.9%)
> | Model                   | Judge Model        | Recall Order | Recall Count | Infer State | Compare State | Predict State | Avg  |
> |-------------------------|--------------------|--------------|--------------|-------------|----------------|---------------|------|
> | Gemini-2.5-Flash        | Qwen2.5-72B        | 44.6         | 41.7         | 27.9        | 27.1           | 13.8          | 31.0 |
> | Gemini-2.5-Flash        | Qwen3-235B-A22B    | 45.8         | 40.8         | 27.1        | 26.2           | 14.2          | 30.8 |
> | Gemini-2.5-Flash w/o think | Qwen2.5-72B     | 22.5         | 34.2         | 19.6        | 20.4           | 8.8           | 21.1 |
> | Gemini-2.5-Flash w/o think | Qwen3-235B-A22B | 22.5         | 32.9         | 19.6        | 20.4           | 9.6           | 21.0 |
> | Seed1.5-VL              | Qwen2.5-72B        | 24.2         | 27.1         | 3.8         | 7.9            | 3.8           | 13.3 |
> | Seed1.5-VL              | Qwen3-235B-A22B    | 23.8         | 25.8         | 2.1         | 6.7            | 3.8           | 12.4 |

---

> ### Author Response · Authors · 2025-11-25
> **Response to Reviewer tQvt (2/5)**
>
> > Continue response to W1
>
> Finally, we provide two illustrative examples of original vs. paraphrased answers alongside their respective judgments by Qwen2.5-72B. Despite noticeable differences in phrasing and formatting, the judge produces consistent and correct evaluations.
>
> ```
> ### Example1
> **Ground-Truth**:
> `1st: (c,2) up\n2nd: (c,1) right\n3rd: (c,2) left\n4th: (c,3) left\n5th: (b,3) down\n6th: (c,3) up\n7th: (b,3) down\n8th: (a,3) down\n9th: (a,2) right\n10th: (b,2) up`
>
> **Original Answer** (Judgement=correct):
> `1st: (c,2) Up\n2nd: (c,1) Right\n3rd: (c,2) Left\n4th: (c,3) Left\n5th: (b,3) Down\n6th: (c,3) Up\n7th: (b,3) Down\n8th: (a,3) Down\n9th: (a,2) Right\n10th: (b,2) Up`
>
> **Paraphrased Answer** (Judgement=correct):
> `1. First: (c,2) moves Up\n2. Second: (c,1) moves Right\n3. Third: (c,2) moves Left\n4. Fourth: (c,3) moves Left\n5. Fifth: (b,3) moves Down\n6. Sixth: (c,3) moves Up\n7. Seventh: (b,3) moves Down\n8. Eighth: (a,3) moves Down\n9. Ninth: (a,2) moves Right\n10. Tenth: (b,2) moves Up`
>
>
> ### Example2
> **Ground-Truth**:
> `(a, 0): black\n(a, 1): black\n(a, 2): black\n(b, 0): black\n(b, 1): white\n(b, 2): black\n(c, 0): white\n(c, 1): white\n(c, 2): black`
>
> **Original Answer** (Judgement=incorrect):
> `(0, a): white, (1, a): white, (2, a): black, (0, b): white, (1, b): black, (2, b): white, (0, c): white, (1, c): black, (2, c): black`
>
> **Paraphrased Answer** (Judgement=incorrect):
> `The final arrangement of the pieces on the grid is as follows: (0, a) and (1, a) are white, while (2, a) is black; (0, b) is white, (1, b) is black, and (2, b) is white; (0, c) is white, (1, c) is black, and (2, c) is also black.`
> ```

---

> ### Author Response · Authors · 2025-11-25
> **Response to Reviewer tQvt (3/5)**
>
> >W2: For Predict Operation, answers are extracted using an LLM, then simulated. This introduces another potential failure point in extraction, but there is no error analysis distinguishing extraction failures from planning failures.
>
> For *Predict Operation*, we follow the same principle for the other five skills. Specifically, we carefully craft task instructions to (1) clarify the valid operations that models are allowed to use and (2) require MLLMs to summarize their final answer in specific formats (Tables 7–12 in the appendix). In this way, the predicted operations are readily extractable using Qwen2.5-72B.
>
> To assess whether extraction introduces additional error, we conducted a manual examination of 200 randomly sampled responses from Gemini‑2.5‑Pro, Gemini‑2.0‑Flash, Seed1.5‑VL and Qwen2.5-VL-7B. In all sampled cases, the extracted operations are consistent with human interpretations of the models’ intended answers. This suggests that the LLM-based extraction is reliable and all failures stem from MLLMs' incorrect answers.
>
> We include several concrete examples below, where the extracted operations produced by Qwen2.5-72B align consistently with the corresponding MLLM outputs
> ```
> **Model Response**
> The final answer is $\boxed{Down, Right, Right}$
> **Extracted Operations**
> ['down', 'right', 'right']
>
> **Model Response**
> No moves are required.
> **Extracted Operations**
> []
>
> **Model Response**
> The sequence of cup swaps to achieve the desired distribution {b1, c1, c2, c3} from the initial state {a1, b1, c2, c3} is: [(a1, c1)]
> **Extracted Operations**
> [('a1', 'c1')]
>
> **Model Response**
> `rm -rf path0/{h.json,k.py,k.txt,u.txt} & touch path0/{c.json,l.csv,n.txt,q.json,r.json,x.py,y.json,z.py}`
> **Extracted Operations**
> ['rm -rf path0/{h.json,k.py,k.txt,u.txt}', 'touch path0/{c.json,l.csv,n.txt,q.json,r.json,x.py,y.json,z.py}']
>
> **Model Response**
> remove 3 of Clubs from pile0, remove 4 of Spades from pile0, remove 7 of Hearts from pile0, remove 9 of Clubs from pile0, add 10 of Clubs to pile0, add 8 of Spades to pile0, add Jack of Spades to pile0, add 3 of Diamonds to pile0
> **Extracted Operations**
> ['remove 3 of Clubs from pile0', 'remove 4 of Spades from pile0', 'remove 7 of Hearts from pile0', 'remove 9 of Clubs from pile0', 'add 10 of Clubs to pile0', 'add 8 of Spades to pile0', 'add Jack of Spades to pile0', 'add 3 of Diamonds to pile0']
> ```

---

> ### Author Response · Authors · 2025-11-25
> **Response to Reviewer tQvt (4/5)**
>
> >W3: vid2txt is generated from the ground-truth state transitions. It works well for isolating perception versus reasoning, but it effectively provides a clean event log the models do not need to perceive. It is possible that the performance gap partly reflects a best-case text abstraction that few real video systems could realistically obtain.
>
> Thank you for this insightful observation. We acknowledge that `vid2txt` provides a "clean event log" rather than a realistic video caption, which may contain noise and irrelevant details.
>
> However, our intention with `vid2txt` is not to simulate real captions, but to serve as an **Oracle ablation study** to decouple reasoning from perception. Providing models with a perfect textual description allows us to probe their reasoning limitations in isolation. We therefore view this setting not as a weakness, but as a necessary diagnostic tool.
>
> This oracle ablation leads to two key findings that would otherwise remain hidden:
> - **Reasoning difficulty exists independently of perception.** Even with perfect event descriptions, non-thinking models perform poorly, and thinking models still struggle on Level 3 Predict problems (Table 2). This indicates that the benchmark’s reasoning demands are substantial on their own.
> - **Current MLLMs lack fine-grained temporal perception**. The notable gap between `vid2txt` and real video performance suggests that fine-grained temporal perception remains a key bottleneck for current MLLMs, leading to a weak foundation for complex video reasoning.
>
>
> >W4: The authors stated that the videos and questions in these benchmarks often resemble those in earlier benchmarks and fall short in demanding deeper video reasoning. It is unclear how this dataset helps address that issue. It is also unclear in what way the previous ones resemble.
>
> Thank you for the thoughtful question. We will address it from two perspectives.
>
> **1. In what way do the benchmarks resemble?**
>
> The two related works referenced in our paper—VCR-Bench [1] and Minerva [2]—primarily aim to evaluate the correctness of intermediate reasoning steps, in addition to the final answer. However, despite this meaningful goal, the underlying skills required to solve their tasks largely resemble those evaluated in prior non-reasoning video benchmarks, such as video temporal grounding [3], event/action counting [4], temporal order understanding [5,6], video knowledge reasoning [7] and motion direction reasoning [5,8]. As a result, many of the tasks can be solved using shallow perception or short-horizon reasoning. For instance, as reported in [2], the average number of words in their annotated reasoning trace is only 92.
>
> Additionally, VCR-Bench and Minerva mainly use videos from common genres (movies, lifestyle, sports, education), which is also similar to existing non-reasoning benchmarks. In fact, VCR-Bench directly reuses videos from earlier datasets.
>
> As we analyzed in Figure 1 and Section 3.3.1 of the paper, previous video understanding benchmarks that focus on the aforementioned skills often lack the reasoning depth necessary to showcase the advantages of extended CoT chains.
>
> **2. How does our benchmark address the issue?**
>
> Our benchmark increases the reasoning depth in two ways:
>
> First, we explicitly represent video content as structured state-transition sequences. This provides a scalable and principled mechanism to adjust complexity—by varying state sizes and operation counts—thereby constructing a solid foundation to assess complex video reasoning.
>
> Second, we introduce three levels of reasoning skills, where Level 2 and Level 3 substantially exceed the difficulty of most existing video understanding benchmarks. These levels require multi-step forward inference or backward tracing to identify the hidden state information (Level 2) and structured planning over long-horizon dependencies to predict the path to future state (Level 3)—capabilities that prior benchmarks do not systematically test.
>
> We also note that [1,2] were released concurrently with our study. Due to anonymity requirements, we cannot provide detailed timing evidence here, but we emphasize that our analyses and benchmark design were developed independently.
>
> [1] VCR-bench: A comprehensive evaluation framework for video chain-of-thought reasoning.
>
> [2] Minerva: Evaluating complex video reasoning.
>
> [3] Hollywood in homes: Crowdsourcing data collection for activity understanding.
>
> [4] Video Question Answering with Spatio-Temporal Reasoning.
>
> [5] TempCompass: Do Video LLMs Really Understand Videos?
>
> [6] Video-MME: The First-Ever Comprehensive Evaluation Benchmark of Multi-modal LLMs in Video Analysis.
>
> [7] Video-MMMU: Evaluating Knowledge Acquisition from Multi-Discipline Professional Videos.
>
> [8] MVBench: A Comprehensive Multi-modal Video Understanding Benchmark.

---

> > ### Author Response · Authors · 2025-11-25
> > **Response to Reviewer tQvt (5/5)**
> >
> > >W5: According to the paper, all videos are synthetic or semi-synthetic demonstrations built from controlled state-transition scripts rather than natural videos. This raises the concern that it may overfit to symbolic reasoning rather than perceptual or intuitive physics reasoning. It may also underestimate challenges such as motion blur, occlusion, object continuity, or ambiguous visual events found in open-world settings. But this is a minor concern. Can you analyze the potential bias introduced by the six procedural video types and clarify whether these discrete settings generalize to naturalistic visual reasoning?
> >
> > We thank the reviewer for the insightful comment. We agree that VideoReasonBench emphasizes discrete, symbolic, and step-by-step reasoning over intuitive physical reasoning or noisy open-world perception. This is a deliberate design choice aimed at probing the current limitations of MLLMs in video reasoning. We would like to address the concern from the following perspectives:
> >
> > **1. Symbolic representation enhances reasoning depth**
> >
> > Our primary motivation is to address the critical gap in reasoning depth within existing video benchmarks (Section 1). While benchmarks like Video-MME are valuable for evaluating open-world visual understanding, models can often succeed using shallow visual cues or prior knowledge without requiring in-depth multi-step reasoning. By representing videos as structured, symbolic state-transition sequences, VideoReasonBench sets a substantially higher bar for reasoning than previous benchmarks (as illustrated in Figure 1 and Figure 5).
> >
> > **2. Symbolic reasoning generalizes to real-world video reasoning problems**
> >
> > While our tasks are abstracted, the cognitive skills they evaluate (temporal ordering, latent state tracking, and operation planning) are fundamental to many real-world video applications. These include:
> > - **GUI navigation**: Tracking and predicting system states based on observed on-screen actions.
> > - **Robotics and Embodied AI**: Planning manipulation sequences by understanding history action and state transision sequences.
> > - **Surveillance Analysis**: Tracking the existence of certain subjects in an environment.
> >
> > While we cannot independently construct a new real-world benchmark to further validate generalization, we provide evidence from an external source. Specifically, we compare the performance of several frontier MLLMs on VideoReasonBench with their rankings on the LMArena vision leaderboard, where models are measured by human preference votes on their response to user-submitted real-world questions. The results below show a consistent correlation, which suggests that VideoReasonBench performance is indeed a strong indicator of a model's broader, generalized visual reasoning skills.
> >
> > **VideoReasonBench vs LMArena Vision performance (Nov 23, 2025)**
> > | Model            | VideoReasonBench | LMArena |
> > |------------------|------------------|---------|
> > | Gemini-2.5-Pro   | 56.0             | 1249    |
> > | Gemini-2.5-Flash | 27.4             | 1214    |
> > | o4-mini          | 10.7             | 1202    |
> > | Gemini-2.0-Flash | 10.4             | 1169    |
> > | GPT-4o           | 6.9              | 1119    |
> >
> > **3. Visual perception in our benchmark is challenging for current MLLMs**
> >
> > Even within our controlled settings, the demand for fine-grained perception poses a significant challenge for today's leading MLLMs. As shown in Table 2, most models struggle significantly even on basic Level 1 (Recall) tasks. If a model cannot reliably track discrete state changes in a clear environment, it is highly unlikely to succeed at more complex scenarios with visual noise and ambiguity of open-world videos.
> >
> > **4. Clarify Scope and Future Work**
> >
> > We agree that "intuitive physical reasoning" (e.g., predicting the trajectory of a falling cloth or fluid dynamics) is a crucial aspect of video understanding. However, we view this as a complementary domain to our focus on **logical and symbolic reasoning**.
> > - **Scope**: VideoReasonBench is designed to be the "GSM8K" or "ARC-AGI" of video understanding—benchmarks that are intentionally abstract to probe reasoning depth—rather than a substitute for general video QA benchmarks.
> > - **Future Work**: We appreciate the suggestion regarding open-world noise. In future iterations, we plan to take this aspect into consideration and incorporate open-world videos to further test the robustness of video reasoning under greater perceptual stress.

---

> > > ### Author Response · Authors · 2025-11-28
> > > **Looking Forward to Your Feedback**
> > >
> > > Dear Reviewer tQvt,
> > >
> > > We sincerely appreciate the time and effort you have invested in reviewing our paper. As the discussion period concludes on December 3rd, we would like to gently check whether our responses have sufficiently addressed your concerns. If any questions remain or if there are points that would benefit from additional clarification, we would be more than happy to provide further details. On the other hand, if our rebuttal has resolved your concerns, we would be truly grateful if this could be reflected in your final evaluation.
> > >
> > > Thank you again for your thoughtful review. We look forward to your feedback.
> > >
> > > Sincerely,
> > >
> > > VideoReasonBench Authors

---

### Official Review · Reviewer_mxvW · 2025-11-03

**Soundness:** 3
**Presentation:** 3
**Contribution:** 3
**Rating:** 6
**Confidence:** 4

**Summary:**

The paper introduces a vision-centric benchmark for complex video reasoning. By revealing latent states only at the beginning or end of the video and presenting a sequence of visible actions, the benchmark systematically evaluates three progressive levels of reasoning: (Level 1) exact recall of the action sequence and counts; (Level 2) inference of invisible states from observed actions and comparison of differences; (Level 3) counterfactual prediction of future states or required actions. The benchmark spans six scenario types (numeric sliders, board flipping, cup–lid swapping, file operations, card stacks, chip–cup), comprising 240 videos and 1440 QA pairs, with evaluation via an LLM-based textual judge and executable simulation.

**Strengths:**

1. The work effectively identifies a key deficiency in current video benchmarks: weak alignment with reasoning and low demand for exploiting video information. For instance, Table 2 shows that using raw video input underperforms a video-to-text pipeline, and Table 3 indicates that prior benchmarks can be partially solved with text-only inputs.
2. As a benchmark for assessing MLLM reasoning, it demonstrates a positive correlation between accuracy and scaled reasoning length, which aligns with expectations for reasoning-focused evaluation.

**Weaknesses:**

1. The task format is relatively narrow, covering only six highly customized scenarios. If models are exposed to similar data during training, they may quickly overfit or “hack” the benchmark.
2. The substantial gap between open-source and closed-source models in Table 2 warrants analysis. Is this driven by differences in training data coverage or by capability gaps? For example, open-source models may have been trained on fewer or less diverse data and thus not encountered similarly customized scenario tasks, leading to lower performance.
3. In Fig. 5(b), the largest thinking budget (8192) yields the shortest response length. Please explain this phenomenon, e.g., by providing illustrative cases.

**Questions:**

N/A

---

> ### Author Response · Authors · 2025-11-25
> **Response to Reviewer mxvW (1/2)**
>
> Dear Reviewer mxvW,
>
> We greatly appreciate your thoughtful comments on our work. We are encouraged by your acknowledgement that our work `effectively identifies a key deficiency in current video benchmarks` and our benchmark `aligns with expectations for reasoning-focused evaluation`. Below we address each of your concerns in detail:
>
> >W1: The task format is relatively narrow, covering only six highly customized scenarios. If models are exposed to similar data during training, they may quickly overfit or “hack” the benchmark.
>
> Thank you for the constructive comment. We agree that expanding video scenario diversity is a valuable future direction of our work. At the same time, we would like to clarify four points:
> 1. VideoReasonBench effectively differentiates existing MLLMs in their ability to perform complex video reasoning and **the observed gaps are not attributed to benchmark hacking**; we provide detailed evidence for this in our response to W2.
> 2. VideoReasonBench is robust against shortcuts such as language priors and static visual cues, as demonstrated by the results in Table 3. This indicates that models cannot "hack" our benchmark by simply memorizing superficial biases.
> 3. **The six video scenarios span three distinct and representative categories**: (i) synthetic, game-style videos (*Number*, *Circle* and *Cup*),  (ii) operating-system–style screen recordings  (*File*) and (iii) real-world videos (*Card* and *Chip*). Together, these cover a broad range of visual dynamics.
> 4. Even with more diverse scenarios, any benchmark can in principle be hacked by constructing targeted training data. As long as future models do not deliberately overfit to our task patterns, we believe that VideoReasonBench will continue to offer effective feedback on their video reasoning ability.
>
> >W2: The substantial gap between open-source and closed-source models in Table 2 warrants analysis. Is this driven by differences in training data coverage or by capability gaps? For example, open-source models may have been trained on fewer or less diverse data and thus not encountered similarly customized scenario tasks, leading to lower performance.
>
> Thank you for the insightful question. While we don not have access to the training data distribution of closed-source models, several pieces of evidence suggest that the observed gap is primarily due to differences in model capability rather than training data coverage:
> 1. **Performance difference across closed-source models.** With the exception of the Gemini-2.5 series, other leading closed-source models also obtain very low accuracy (<12%). It is unlikely that only Gemini-2.5 models were uniquely exposed to tasks resembling VideoReasonBench, while the other frontier systems were not. This pattern is more consistent with inherent capability differences.
> 2. **Intra-family discrepancies under similar training data.** Models within the same family—where training data is generally expected to be highly similar—exhibit substantial performance gaps. For example, Gemini-2.5-Pro outperforms Gemini-2.5-Flash by 28.6%, and Qwen2.5-VL-72B achieves 7.2% accuracy compared to Qwen2.5-VL-7B’s near-zero 1.3%. These discrepancies point to capability scaling effects rather than data exposure differences.
> 3. **Within-model improvements due to “thinking” mode.** Even for the same model, Gemini-2.5-Flash shows an 8.6% accuracy gain when using its thinking mode. Since training data coverage is identical across modes, this improvement highlights the role of enhanced reasoning ability, not differences in data.
> 4. **Positive correlation with LMArena performance.** As shown in the Table below, the performance of several SOTA MLLMs on our benchmark exhibits a positive correlation with the LMArena vision leaderboard, where models are measured by human preference votes on their response to user-submitted real-world questions. This alignment indicates that strong performance on VideoReasonBench reflects broader visual understanding and reasoning capabilities, rather than superficial pattern memorization from training data.
>
> **VideoReasonBench vs LMArena Vision performance (Nov 23, 2025)**
> | Model            | VideoReasonBench | LMArena |
> |------------------|------------------|---------|
> | Gemini-2.5-Pro   | 56.0             | 1249    |
> | Gemini-2.5-Flash | 27.4             | 1214    |
> | o4-mini          | 10.7             | 1202    |
> | Gemini-2.0-Flash | 10.4             | 1169    |
> | GPT-4o           | 6.9              | 1119    |

---

> > ### Author Response · Authors · 2025-11-25
> > **Response to Reviewer mxvW (2/2)**
> >
> > >W3: In Fig. 5(b), the largest thinking budget (8192) yields the shortest response length. Please explain this phenomenon, e.g., by providing illustrative cases.
> >
> > Thank you for highlighting this interesting observation. As shown in Fig. 5(b), the number of thinking tokens increases steadily as the thinking budget grows, yet the number of response tokens (i.e., summary tokens) drops noticeably at the largest budget (8,192). We conjecture that this occurs because, when the thinking budget is limited (e.g., ≤2,048), the model cannot complete the full reasoning process within the “thinking” phase. As a result, it is forced to continue part of the reasoning in the response phase, which leads to longer and more verbose responses. In contrast, with a sufficiently large thinking budget (8,192), the model can finish most of its reasoning in the “thinking” phase, allowing the final response to remain concise and focused.
> >
> > To illustrate this effect, below we compare the responses generated by Gemini-2.5-Flash under thinking budgets of 8,192 and 2,048. Under the 8,192 setting, the model produces a direct and succinct step-by-step answer. Under the 2,048 setting, however, the response contains redundant segments, including a “re-examination” step where the model corrects earlier reasoning errors within the response itself. This comparison supports our conjecture that **insufficient thinking budget pushes the model to shift part of the reasoning into the response, thereby increasing response length**.
> >
> >
> > **Question**
> > ```
> > What is the arrangement of numbers on the board at the start of the video?
> > ```
> >
> > **Gemini-2.5-Flash (think budget=8,192; thinking token=6,907; response token=1,010)**
> > ```
> > ...
> > The final state is:
> > ...
> > The sequence of moves shown by the arrows is:
> > ...
> > To reverse, we start from the final state
> > ...
> > Reverse Move 6 (reverse of (c,3) -> (c,2)):
> > ...
> > Reverse Move 1 (reverse of (b,3) -> (a,3)):
> > ...
> > The arrangement of numbers on the board at the start of the video is:
> > ...
> > ```
> >
> > **Gemini-2.5-Flash (think budget=2,048; thinking token=2,566; response token=2,878)**
> > ```
> > At the end of the video (00:13) ... The empty square is at (c,3). The numbers are:
> > ...
> > To find the initial arrangement, we must reverse the moves shown in the video. The moves were:
> > ...
> > Let's reverse these steps starting from the final state:
> > ...
> > Reverse Move 5 (Empty moves from (c,3) to (c,2)
> > ...
> > Reverse Move 1 (Empty moves from (a,3) to (b,3)
> > ...
> > This derived initial state has the empty square at (b,3), but the video clearly shows the empty square starting at (a,3). Let's re-examine the first move.
> > ...
> > Okay, re-watching the video frame by frame
> > ...
> > This initial state has the empty square at (a,3), which matches the video. Let's list the coordinates and numbers.
> > ...
> > ```

---

> > > ### Author Response · Authors · 2025-11-28
> > > **Looking Forward to Your Feedback**
> > >
> > > Dear Reviewer mxvW,
> > >
> > > We sincerely appreciate the time and effort you have invested in reviewing our paper. As the discussion period concludes on December 3rd, we would like to gently check whether our responses have sufficiently addressed your concerns. If any questions remain or if there are points that would benefit from additional clarification, we would be more than happy to provide further details. On the other hand, if our rebuttal has resolved your concerns, we would be truly grateful if this could be reflected in your final evaluation.
> > >
> > > Thank you again for your thoughtful review. We look forward to your feedback.
> > >
> > > Sincerely,
> > >
> > > VideoReasonBench Authors

---

### Author Response · Authors · 2025-11-25
**General Response**

We sincerely thank all reviewers for their comprehensive review and valuable feedback on our paper.

We are encouraged by the positive comments highlighting the strengths of our work: that it is `well motivated` (4Dkk), `effectively identifys a key deficiency in current video benchmarks` (mxvW), and that `our formulation of videos as  latent state + visible operations makes failure modes inspectable and comparable` (qkUM). Reviewers also noted that our benchmark `aligns with expectations for reasoning-focused evaluation` and offers `reasonable size and balanced distributions` (tQvt and 4Dkk).

We address all concerns and questions in detail in the point-by-point responses to each reviewer. In addition, we have revised the manuscript according to the constructive suggestions provided. For ease of reading, all updated sections are highlighted in blue. Below, we provide a brief summary of the major revisions:

- Added consistency analysis of LLM-as-Judge in Appendix A.1. (Reviewer tQvt W1, qkUM W1)
- Included Qwen2.5-VL-7B/72B results under the `vid2txt` setting in Table 2. (Reviewer tQvt W3, Reviewer 4Dkk W2)
- Added comparison between VideoReasonBench and LMArea Vision Leaderboard performance in Appendix A.2. (Reviewer tQvt W5, mxvW W2)
- Added analysis on further extending the state size and operation count in Appendix A.3. (Reviewer qkUM W3)
- Expanded discussion of two concurrent work—VCR-Bench and Minerva—in the Introduction, to clarify how they relate to prior general video understanding benchmarks. (Reviewer tQvt W4)
- Added discussion in Section 3.2, emphasizing that the gap between Gemini-2.5 and other models is not due to benchmark "hacking". (Reviewer mxvW W2)
- Added a seperate **Limitation and Future Work** section describing our plan to incorporate open-world videos with greater perceptual challenge in future work. (Reviewer tQvt W5)

We are grateful for the reviewers’ insights and believe these revisions have significantly strengthened the paper. If the reviewers have any further questions, we are more than happy to discuss.

---

### Author Response · Authors · 2025-11-29
**Rebuttal Summary**

Dear PCs, ACs and Reviewers,

Thank you again for your time and effort in reviewing our paper. We have carefully addressed each comment and provided a point-by-point response to each reviewer. For your convenience, we summarize the key points below:

# Review Summary
We are encouraged by the positive feedback from initial reviews. Three reviewers rated our work as **good** across *Soundness*, *Presentation*, and *Contribution*, and one reviewer rated the *Soundness* as **excellent**. Reviewers highlighted several strengths of our work, including:
- The paper is well motivated (4Dkk), effectively identifys the weak alignment with reasoning and low demand for exploiting video information in current video benchmarks (mxvW).
- The task design indeed requires visual reasoning to solve (4Dkk) and the results align with expectations for reasoning-focused evaluation (mxvW), making failure modes inspectable and comparable (qkUM).
- The benchmark offers reasonable size and balanced distributions (tQvt and 4Dkk).

# Response to Key Questions & Concerns
>1. Robustness of using LLM-as-Judge in our evaluations. (tQvt W1, QkUM W1)

The reviewer is concerned about the consistency and correctness of using LLM in our evaluations.

To quantify the robustness of LLM-as-Judge, we conducted additional experiments to confirm that evaluation is consistent with respect to both answer paraphrases and judge model choices. We also manually review a sampled set of the LLM judgement results and find that they match our own assessments. We believe these results provide compelling evidence for the validity of our evaluations.

>2. The potential risk of models overfitting or hacking our benchmark. (mxvW W1)

The reviewer is concerned about the potential for models to overfit or "hack" our benchmark due to  perceived narrowness of video scenarios.

- We provide several concrete evidences showing that the performance gap between current MLLMs are not due to "hacking" the benchmark but reflect genuine video reasoning capabilities.
- We clarify that videos in our benchmark are designed with diversity, incorporating three distinct categories: (i) synthetic, gaming-style videos, (ii) operating-system–style screen recordings and (iii) real-world videos.
- We show that our tasks are robust against shortcuts such as language priors and static visual cues (Table 3).

>3. Generalization of symbolic reasoning to open-world videos. (tQvt W5)

The reviewer is concerned that our video content emphasizes symbolic reasoning skills and might fail to generalize to open-world video reasoning problems.

- We clarify that the emphasis on symbolic and step-by-step reasoning is a deliberate design choice to address the lack of reasoning depth in current video understanding benchmarks.
- We argue that the reasoning skills tested (e.g., temporal ordering, latent state tracking, and operation planning) are fundamental to real-world video applications like GUI navigation and robotics.
- We provide new evidence showing a strong correlation between model performance on our benchmark and their rankings on the LMArena vision leaderboard, suggesting that success in our tasks is a good indicator of broader, generalized visual reasoning abilities.

>4. The main cause of MLLM failures in our benchmark. (4Dkk)

The reviewer questions if the performance gap in the vid2txt setting (Table 2, where we replace raw videos with structured text representations and observe a notable boost in model performance) implies that the primary challenge in our benchmark is perception rather than reasoning.

We clarify that while fine-grained temporal perception is a bottleneck, complex reasoning remains a substantial and independent challenge in our benchmark.
- We provide strong evidence from our vid2txt results showing that even with perfect text descriptions, "non-thinking" models still perform very poorly, lagging behind "thinking" models by a significant margin (30-50%).
- We further highlight that even advanced "thinking" models struggle with the most complex reasoning tasks (Level 3 Predict) in the vid2txt setting.

These findings demonstrate that our benchmark effectively evaluates both fine-grained perception and complex reasoning.

# Discussion Summary
- Reviewer 4Dkk acknowledged our rebuttal and maintained the positive assessment.
- Due to the early closure of the discussion phase, the remaining reviewers did not provide follow-up comments.

---

### Meta-Review · Area_Chair_RSGT · 2026-01-11

**Summary:**

Reviewers appreciated the proposed benchmark (e.g., benchmark size, balanced distirbution, challenging nature) and comprehensive evaluation covering a wide span of open-source MLLMs. Reviewers also raised some concerns in the initial review including, employing LLM-as-Judge in experimental evaluation, generalization of symbolic reasoning to open-world videos, overfitting issues on the benchmark, additional analysis with respect to the gap between open-source and closed-source models in Table 2, and analysis regarding scenarios comprising minutes-scale durations.

**Reviewer Concerns:**

Authors provided a comprehensive rebuttal to address the initial concerns raised by the reviewers. For instance, authors provided additional analysis to analyze the robustness of LLM-as-Judge (e.g., paraphrasing the MLLMs' responses and comparing the evaluation results to that of the original answers, employign a different judge model (Qwen3-235B-A22B), suggesting that the results remained consistent. In additon, authors also manually reviewed 200 randomly sampled responses with 50 samples from each of the four models (Gemini‑2.5‑Pro, Gemini‑2.0‑Flash, Seed1.5‑VL and Qwen2.5-VL-7B), with the results correlating with the LLM judgement results. Authors presented additional analysis to study the risk of models overfitting. Authors provided additional results suggesting a correlation between model performance on their proposed benchmark and their rankings on the LMArena vision leaderboard. Authors extended the study to analyze the different causes of MLLM failures on the proposed benchmark, suggesting that both fine-grained temporal perception and reasoning pose challenges to achieving better performance. Post-rebuttal, one of the reviewers mentioned to be satisfied with the authors response, keeping the positive rating.

**Reviewer Scores:**

The meta-reviewer believes most of the concerns of the the reviewers were addressed in the rebuttal. Three of the reviewers are generally on the positive side. For the remaining reviewer, authors provided a detailed response such as addressing consistency of LLM-as-judge across paraphrased or differently formatted answers. Given that most of the reviewers are on the positive side, a comprehensive rebuttal provided by the authors, the meta-reviewer believes the strength outweights weaknesses for this manuscript and agrees with the majority of the reviewer's positive ratings.

---

### Decision · Program_Chairs · 2026-01-26

Accept (Poster)